# Current Status of Indole-Derived Marine Natural Products: Synthetic Approaches and Therapeutic Applications

**DOI:** 10.3390/md22030126

**Published:** 2024-03-06

**Authors:** Sergio Fernández, Virginia Arnáiz, Daniel Rufo, Yolanda Arroyo

**Affiliations:** 1Department of Chemistry, School of Physical and Chemical Sciences, Queen Mary University of London (QMUL), Mile End Road, London E1 4NS, UK; s.f.gonzalez@qmul.ac.uk; 2Department of Organic Chemistry, Science Faculty, University of Valladolid (UVa), Paseo de Belén 7, 47011 Valladolid, Spain; virginia.arnaiz22@uva.es (V.A.); daniel.rufo@estudiantes.uva.es (D.R.); 3Department of Organic Chemistry, ITAP, School of Engineering (EII), University of Valladolid (UVa), Dr Mergelina, 47002 Valladolid, Spain

**Keywords:** indole alkaloids, marine resources, biological activity, therapeutic application, synthetic strategies

## Abstract

Indole is a versatile pharmacophore widely distributed in bioactive natural products. This privileged scaffold has been found in a variety of molecules isolated from marine organisms such as algae and sponges. Among these, indole alkaloids represent one of the biggest, most promising family of compounds, having shown a wide range of pharmacological properties including anti-inflammatory, antiviral, and anticancer activities. The aim of this review is to show the current scenario of marine indole alkaloid derivatives, covering not only the most common chemical structures but also their promising therapeutic applications as well as the new general synthetic routes developed during the last years.

## 1. Introduction

Marine organisms constitute an important source of natural products with tremendous biological and chemical diversity. Sponges, algae, corals, marine bacteria, and fungi were shown to produce new secondary metabolites that may be the key to the production of new drugs to treat various diseases [1,2]. In this regard, marine natural products have important advantages over those of terrestrial origin, including chemical novelty, new mechanisms of action, and greater biological activity. These valuable pharmacological properties can be explained due to the fact that many marine compounds have evolved to fight for their organism survival, becoming very powerful inhibitors of biological processes in the predators of the marine organisms that utilize them for survival. [3]. The anticancer drugs Trabectedin (Yondelis^®^; Figure 1) and Eribulin mesylate (Halaven^®^; Figure 1) are examples of marine drugs accepted by the FDA that proceed by a novel mechanism of action [4,5]. On the other hand, the cyclic depsipeptide Largazole, isolated from a cyanobacterium, is one of the most potent class I histone deacetylase inhibitors, and the first cyanobacterial secondary metabolite containing a thioester (Figure 1) [6,7].

As stated before, marine organisms have proven to be an outstanding source of active molecules, with indole derivatives being one of the most promising [8]. Chemically, indole (2,3-benzopyrrole) consists of benzene and pyrrole rings fused together. Indole is an important industrial product widely used in the production of fragrances [9], medicines [10], exogenous auxins [11], and colorants like indigo. The indolyl group is an important fragment present in a wide variety of natural products, such as the amino acid Tryptophan (Trp), which is involved in the synthesis and release of the neurotransmitter serotonin (related to mood), the hormone melatonin (which regulates sleep), indole alkaloids, and the plant hormone auxin. Therefore, this moiety has also received much attention in the fields of synthetic organic chemistry and medicinal chemistry [8]. Importantly, recent research has shown clear evidence of the relationship between the chemical structure of the indole bicyclic skeleton and the biological activity it presents. In this sense, anticancer [12,13,14], anti-coronavirus [15,16], and anti-diabetic [17,18,19,20] activities are observed when there are amide or chalcone groups at the C2 and/or C3 positions of the indolyl group. Anti-Alzheimer’s disease activity [21] is observed when seven-membered nitrogen-containing heterocycles are present at the C2 and/or C3 positions. Anti-inflammatory [22,23] and antifungal activities [24,25] are observed when different functional groups are placed at the N1 position. Finally, inhibition against osteoporosis [26] is observed when a thiophene group is installed at the C7 position (Figure 2).

Recently, Martinez et al. described several marine natural products as Breast Cancer Resistance Protein (BCRP) inhibitors [27]. Among them, three examples stand out: Fumitremorgin C (FTC), a prenylated indole alkaloid derived from the amino acids L-tryptophan and L-proline; Tryprostatin A, a natural analog of FTC formed by the condensation of a proline unit and an isoprenyl tryptophan residue into a diketopiperazine unit; and the β-carboline alkaloid Harmine (Figure 3). It is noteworthy to mention that all these compounds are indole alkaloids.

Undoubtedly, many compounds derived from marine sources have marked milestones in the treatment of diseases. In particular, indole-containing alkaloids, one of the largest, most abundant, and most chemically diverse family of natural compounds, have been shown to have outstanding potential in the development of new drug leads. However, there are still many obstacles to overcome, in particular the devastating side effects and the fact that many cancers develop resistance to several important pharmaceuticals. For these reasons, it is necessary to continue searching for new, safer, and more efficient drugs. Within this context, the structural and functional versatility of indole alkaloid derivatives spots them as privileged scaffolds that could streamline the discovery of chemical analogs with potential applications in drug discovery. Therefore, the purpose of this review is to exclusively cover indole alkaloid derivatives from marine sources with a therapeutic interest, as well as the novel synthetic routes described to obtain these versatile compounds. The relationship between their chemical structure and bioactivity is also addressed in those cases that are described in the literature. 

## 2. Marine Indole Alkaloids

Marine Indole Alkaloids (MIAs) present many different structural features and exhibit wide biological activities such as anti-inflammatory, anticancer, anti-HIV, antibacterial, antifungal, and anti-diabetes activity, among many others. Both aspects require being organized and ordered. In this section, the origin and therapeutic applications of MIAs are presented. Furthermore, synthetic routes from a large number of MIA families have also been included. Based on chemical structures, indole alkaloids can be classified into three groups: simple indole alkaloids, prenylated indoles, and annelated indoles.

### 2.1. Simple Indole Alkaloids, (SIAs)

Simple indole alkaloids consist of an indole nucleus with distinct substitution patterns at the N1, C3, C4, C5, C6, C7, and C8 positions [28,29]. In this section, the compounds of this group are ordered according to the complexity of the substituents of the indole moiety, starting from acyclic to cyclic ones. 

#### 2.1.1. C3-Acyclic Substituted Simple Indole Alkaloids

The most common substitution in simple indole alkaloids occurs at the C3 position, a characteristic observed in many families of simple alkaloids [8,30]. Various examples showcase the biological activities of these compounds (Figure 4). For instance, tryptophol (2-(1*H*-Indol-3-yl)ethan-1-ol, **1**) isolated from the marine sponge Ircinia spiculosa, showed sleep-inducing activity [31]. On the other hand, 2-(1*H*-indol-3-yl)ethyl 2-hydroxypropanoate (**2**), isolated from the yeast strain (USF-HO25) of a marine sponge identified as *Pichia membranifaciens*, exhibits a mild response as a radical scavenger of 2,2-diphenyl-1-picrylhydrazyl (DPPH) [32]. Another example is methyl 1*H*-indole-3-carboxylate (**3**), obtained from *Spongosorites* sp., a marine sponge, demonstrating cytotoxic attributes against several human cancer cell lines [33]. Additionally, Hainanerectamine B (**4**), isolated from *Hyrtios erecta*, a marine sponge from Hainan, has shown the ability to inhibit Aurora A, a serine/threonine kinase involved in the regulation of cell division [34]. Finally, Tryptamine (**5**), was obtained from *Fascaplysinopsis reticulata*, a lyophilized sponge, and demonstrated antibacterial and growth inhibition activity towards *Vibriocarchariae* (MIC = 1 μM) [35].

The presence of carbonyl or carboxyl groups in the indole ring has demonstrated different and interesting biological activities [28,36,37]. Compound **6**, an indole carbaldehyde obtained from *E. chevaleri* KUFA 0006, a culture of an endophytic fungus, exhibited inhibitory activity against *S. aureus* ATCC 2592 biofilm settlement [38]. Hytiodoline (**7**) an indole amino acid obtained from the *Hyrtios* sponge, demonstrated potent anti-trypanosomal activity [39]. Becillamide A (**8**), a thiazole indole derivative obtained from *Bacillus* sp. marine bacterium, showed antibiotic activity against *Archangium gephyra*, immunosuppressing the myxobacterium [40]. Anthranoside (**9**), containing a carboxylated aniline, was obtained from the sponge-originated actinomycete, *Streptomyces* sp. CMN-62, and exhibited anti-influenza activity against the H1N1 virus and inhibited the reaction to NFκB [41]. Hermanine D (**10**), isolated from ascidian *Herdmania momus*, could inhibit the mRNA expression of iNOS, consequently provoking an anti-inflammatory effect [37] (Figure 5).

Quia Che and coworkers developed a biogenetic route to obtain Anthranoside C (**11**), starting with anthranilic acid (**12**) and D-glucose (**13**). The process involves linking two **12** molecules to a **13** molecule, to create a benzenaminium salt; the subsequent cyclization creates the indole ring and, in one further step, the Anthranoside C (**11**) [41] (Figure 6).

The prenylated simple indoles have also demonstrated diverse and fascinating biological activities [28,36,37]. In this sense, the prenylated indole carbaldehyde (**14**), obtained from *E. chevaleri* KUFA 0006, exhibited inhibitory activity against *S. aureus* ATCC 2592 biofilm settlement [37]. Eurotiumin (**15**), an amide indole derived from *Eurotium* sp. SCSIO F452, a sediment-derived fungus from the South China Sea [42], showed antioxidant properties in a DPPH assay [43]. Misszrtine A (**16**), an unusual *N*-substituted prenylated indole obtained from *Aspergillus* sp. SCSIO XWS03F03, a sponge-derived fungus, exhibited strong activity against HL60 and LNCaP cell lines [44]. Terpetin (**17**), a polyamide indole obtained from *Aspergillus* sp. SpD081030G1f1, acted as a protector against L-glutamate toxicity in cells [45] (Figure 7).

May Zin et al. proposed the biogenesis of isomer compounds **18** and **19**, starting with L-tryptophan (**20**). Isomer **17** was obtained in five steps, and isomer **19** required one additional isomerization step (Figure 8) [38].

The compounds described above have underscored the potential of indole alkaloids in both organic and medicinal chemistry, emphasizing the importance of exploring synthetic pathways to obtain simple indole alkaloids. Some straightforward methods for obtaining functionalized simple indole alkaloids include the Bartoli reaction, which involves nitrobenzene (**21**) and vinylmagnesium bromides (**22**) [46]. Another reaction involves the intramolecular Rh-catalyzed decomposition of *ortho*-azydostyrenes (**23**), followed by C–H activation [47]. Additionally, two novel and high-yielding Au(I)-catalyzed reactions have been reported: one involving the intramolecular cyclization reaction of *ortho*-alkynylanilines (**24**) [48] and the other involving the reaction between alkynyl-hydroxycyclohexadienones (**25**) and primary amines [49] (Figure 9).

#### 2.1.2. C3 (Iminoimidazolidin and Pyrazin)-Substituted Simple Indole Alkaloids

Usually, simple indole alkaloids are categorized into families based on similar structures, activities, or origins [36]. Some examples of this classification include Trachycladindoles and Aplysinopsin, both of which feature an iminoimidazolidine ring in the above position (Figure 10). Trachycladindoles A (**26**), C (**27**), G (**28**), B (**29**), D (**30**), E (**31**), and F (**32**), isolated from the marine sponge *Trachycladus laevispirulifer*, have demonstrated cytotoxicity against human cancer cells (HT-29, A549, and MDA-MB-231) [50]. Additionally, Aplysinopsin (**33**) and its derivatives **34**–**40** were obtained from Thorectidae sponges (*Thorectandra* and *Smenospongia*) [51]. They demonstrated activity against *Staphylococcus epidermidis*, with derivative **38** exhibiting the highest antimicrobial activity (MIC = 33 μM). Following this, derivatives **36** (MIC = 36.5 μM), **35** (MIC = 74.6 μM), **34** (MIC = 98.3 μM), and **37** (MIC = 273.8 μM) showed decreasing levels of antimicrobial activity [36]. Derivative **34** was discovered in the Jamaican sponge *Smenospongia aurea*, and it exhibited a high affinity for two receptors, 5-HT2A and 5-HT2C [52]. The latest derivatives, **39**–**40**, were discovered in the marine sponge *Fascaplysinopsis reticulata*. They exhibited remarkable activity against the bacterium *Vibrio natrigens*.; derivative **39** demonstrated potent activity with a MIC of 0.03 μM, while derivative **40** exhibited significant activity (MIC = 2.4 μM) [35].

A hypothetical method for the biosynthesis of trachycladindoles has been described by A. Hentz. The route starts with tryptophan (**20**), and trachycladindole A–G (**26**–**32**) is obtained in 5 steps [53] (Figure 11). 

The synthesis of the Aplysinopsin derivate **39** is shown in Figure 12 and was described by Stanovnik and Svete [54]. The key step for the formation of the iminoimidazolinone core was achieved by the addition of methylamine to a carbodiimide intermediate, followed by an intramolecular amidation reaction.

Meridianins A–G (**41**–**47**) are a family of SIAs characterized by having pyrazine rings at the C3 position (Figure 13). Meridianins are obtained from several sources, but mainly tunicates. Thus, the first was *Aplidium meridianum*, from which Meridianins A–E were isolated, [55] but other examples include *Aplidium falklandicum* and *Synoicum* sp. [56]. 

Meridianins B–E (**42**–**45**) are notable for their demonstrated cytotoxic effects against adenocarcinoma and murine mammary tumor cell lines (IC_50_ = 11.4, 9.3, 33.9, and 11.1 μM, respectively) [55]. Moreover, **44** exhibited antibiofilm potential against methicillin-resistant *Staphylococcus aureus* (MRSA) [36]. In general, the Meridianin family demonstrated wide biological activities which are summarized in Table 1 [56,57]. 

Meridianins can be formed through several synthetic routes, the first one developed by Jiang and Yang from a 7-bromoindolylboronic acid (**48**) and a 4-chloro-pyrimidinyl-2-amine (**49**). In only two steps, **44** was obtained with a high yield [58] (Figure 14). 

However, Fresneda and Molina’s methodology to obtain Meridianins **43** and **44** is the most used route to date. Starting from the corresponding brominated indoles, this four-step route presents high yields in all reactions [59] (Figure 15). 

It is noteworthy to mention that most methodologies focus on the synthesis of **43** and **44** [58]. For example, Karpov et al. improved a three-component palladium-catalyzed carbonylated alkylation [60], while Müller and coworkers achieved it in a one-pot procedure based on Suzuki coupling following a Masuda borylation with a palladium catalyst [61]. Zhou and Chen developed a route to **43** and its derivatives [57], and Penoni employed the indolization of nitrosoarenes to obtain **43** derivatives with the indole moiety functionalized [62]. Remarkably, Stanovnik and Svete described the synthesis of Meridianins **41**–**47** and the Aplysinopsins derivatives **33**–**40** [54]. 

#### 2.1.3. Bis-/Tri-Indole Alkaloids

Bis- and tris-indole alkaloids are characterized by the linkage of the indole moieties through (hetero)carbonated chains, typically between the C3 positions [36]. Usually, when indole alkaloids are bridged by an imidazole ring, they exhibit interesting biological activity. Examples of such cases are bis-indoles Dihydrospongotine C (**50**), Spongotine C (**52**), and the tris-indole Tulongicin (**54**), isolated from the sponge *Topsentia*. They have demonstrated antiviral activity against HIV, HxB2, and YU2, with IC_50_ values ranging from 2.7 to 12 μM and 3.5 to 9.5 μM, respectively, as well as antimicrobial and antibacterial properties, particularly against *S. aureus* (MIC = 1.8 to 7.6 μM) [63] (Figure 7). Furthermore, Rhopaladin C (**53**), isolated from a marine tunicate, demonstrated antimicrobial efficacy against *Sarcina lutea* and *Corynebacterium xerosis* (IC_50_ = 36.9 μM) [64]. Lastly, Spongotine A (**51**) was isolated from the *Topsentia pachastrelloides* sponge and showed antibacterial effects against both the susceptible and methicillin-resistant strains of *S. aureus* [65] (Figure 16). 

Bromodeoxytopsentin (**55**) and dibromodeoxytopsentin (**56**) feature an unsaturated imidazole bridging the indole moiety (Figure 17). Compound **55**, isolated from the *Topsentia pachastrelloides* sponge, demonstrated antibacterial effects against both the susceptible and methicillin-resistant strains of *S. aureus* [65]. Compound **56**, obtained from a genus of the marine sponge *Topsentia*, also exhibited antibacterial properties against *S. aureus* (MIC = 22.7 μM) and showed additional potential as an antiviral agent against HIV (YU2, IC_50_ = 57 μM) [63].

Eusynstelamides A–B (**57**–**58**) and D–F (**59**–**61**) are brominated bis-indoles bridged by a γ-lactam ring obtained from ascidians [66] and bryozoans [67] (Figure 18). Compounds **57** and **58** displayed only weak effectiveness against *S. aureus* [66]. However, compounds **59**–**61** proved stronger antibacterial properties, showing activity against *S. aureus* and *Corynebacterium glutamicum* (MIC = 7.8–17.4 μM), as well as *E. coli* and *P. aeruginosa* (MIC = 16.4–34.7 μM) [67]. 

Hamacanthins A–B (**62**–**63**) are bis-indole isomers linked by a pyrazinone ring, isolated from a marine sponge belonging to the genus *Hamacantha* (Figure 19). Both exhibited antimicrobial properties and efficacy against *B. subtilis* (MIC = 6.4 and 3.3 μM respectively) [68].

Regarding the synthesis of bis- and tris-indole species, an illustrative example could be the synthesis of Rhopaladin C (**53**), developed by Janosik et al. [64]. Starting from 1*H*-indole-3-carbonyl cyanide, the desired product could be obtained by condensation of the nitrile group with the amino group from the L-Tryptophan methyl ester to generate the imidazolone core. This transformation yields **53** in two steps with a moderate yield (Figure 20). 

### 2.2. Prenylated Indole Alcaloids (PIAs)

Prenylated indoles include several different families of compounds. For better insight for the readers, the PIAs are arranged in ascending order of complexity, ranging from an indole core with a cyclic prenyl substituent to an indole moiety fused with a variable number of prenyl-derived ring systems. PIAs containing the indole core with acyclic prenyl substituents are included in Section 2.1.

#### 2.2.1. Diketopiperazine (DKP) Indole Alkaloids

##### Simple Diketopiperazines

Simple 2,5-diketopiperazines (2,5-DKPs) are cyclodipeptides formed through the condensation of two α-amino acids, establishing two amide linkages to form the six-membered ring [69,70]. This kind of compound demonstrated a good catalytic performance in the asymmetric synthesis of the Reformatsky reaction [71]. Furthermore, they have been used as structural fragments in the design of novel drugs [53]. 

Based on their chemical structure, simple indole diketopiperazines include a wide variety of families of compounds [71]. Then, DKPs, that have been found to have biological activities, have been ordered by increasing structural complexity into two main groups: monoindole and bisindole DKPs. Further, monoindole DKPs differ in how the indole is attached to the diketopiperazine, being ultimately divided by the attachment at C3 with a methylene or ethylidene bridge (Figure 21). 

##### Attached at C3 with a Methylene Bridge

This classification has been organized according to the monoindole diketopiperazines containing a diketopiperazine ring attached at the C3 indol core with a methylene bridge (Figure 22 and Figure 23). These indole DKPs are commonly isolated from fungi, such as Aspergillus, and Penicillium, among others [72]. An example of a marine bioactive indole diketopiperazine alkaloid is Brevianamides, which originated from tryptophan and proline. 

The simplest member of this family of compounds is the (S)-Brevianamide F (**64**), derived from the hexahydropyrrolopyrazine and is a precursor of a variety of more complex prenylated alkaloids. Compound **64**, isolated from the marine-derived *Penicillium vinaceum*, showed antibacterial activities against Bacille Calmette-Guérin (BCGs) (IC_50_ = 44.1 µM) and S. aureus, with antifungal activity against *C. albicans* [73,74]. (R)-Cyclo(D-Trp–L-Pro) (**65**), the enantiomer of Brevianamide F (**64**) isolated from the fungi, showed antimicrobial activities [75]. Compound **66**, derived from the fungus Aspergillus fumigatus, bears an N-tert-butoxycarbonyl protecting group which increases its antimicrobial activity against *S. aureus* and *B. subtilis* (IC_50_ = 2.1–3.3 μg/mL) [76].

Another two examples, whose structures derived from the hexahydropyrrolopyrazine, are 18-Oxotryprostatin A (**67**) and compound **68**, both isolated from the marine-derived fungus *Aspergillus sydowi*. 18-Oxotryprostatin A (**67**) exhibited weak cytotoxic activity against A-549 cells (IC_50_ =1.28 µM) [77]. Compound **68** showcased antimicrobial activity against *S. aureus* and *B. subtilis* (IC_50_ = 2.1–3.3 μg/mL). This activity was strongly enhanced due to the C2-isoprene and N-tert-butoxycarbonyl units [76]. 

8,9-Dihydrobarettin (**69**), a brominated cyclodipeptide found in the boreal sponge Geodia barretti, exerted inhibitory activity against AChE and BChE, and potent antifouling, antioxidant, and anti-inflammatory activities, making it a potential lead compound in the prevention of chronic inflammatory diseases [78,79]. Further, it displayed a high affinity for the 5-HT receptor [79,80].

Cyclo(L-Trp–L-Ala) (**70**), Rubrumlines F (**71**), G (**72**), J (**73**), M (**74**), N (**75**), and O (**76**), 5-Piperazinedione **77**, 2,5-piperazinedione **78,** and Preechinulin (**79**), found in the marine-derived fungus *Eurotium rubrum*, demonstrate an effect against the influenza virus strain A/WSN/33 (H_1_N_1_) [81]. Echinulin (**80**), extracted from the marine-derived fungus *Eurotium rubrum* MPUC136, presents two isoprene units in the indole core and showed inhibitory activity against B16 melanoma cells [81,82]. 

The diketopiperazine **78**, obtained from the M-3 strain belonging to the *Ascomycota phylum*, exhibited strong and selective antifungal activity against *Pyricularia oryzae*. 

14-Hydroxyterezine D (**81**) was derived from *Aspergillus sydowi* PFW1-13 and showed weak cytotoxic activity towards A549 (IC_50_ = 7.31 µM). Further, it was active against HL-60 cells (IC_50_ = 9.71 µM) [77]. Didehydroechinulin (**82**) was isolated from *Eurotium cristatum* EN-220 and showed potent lethal activity against brine shrimp and a weak nematicidal effect against *Panagrellus redivivus* (IC_50_ = 27.1 μg/mL) [83]. Both have one and two isoprene units in the indole core respectively.

##### Attached at C3 with an Ethylidene Bridge

Isoechinulin B (**83**), Cryptoechinuline G (**84**), and alkaloid E-7 (**85**) have been isolated from the marine-derived fungus *Eurotium rubrum* MPUC136 [84], and feature several isoprene units in the indole core. They exhibited inhibitory activity against melanin synthesis using B16 melanoma cells [81,82]. Demethyl-12-oxo-eurotechinulin B (**86**), obtained from the same fungal strain, showed cytotoxic activity against the SMMC-7721 cell line (IC_50_ = 30 μg/mL) [43] (Figure 24). 

Cristatumin A (**87**), isolated from *Eurotium cristatum* EN-220, showed antibacterial activity against *S. aureus* and *E. coli* (IC_50_ = 64 and 8 μg/mL). As far as we can ascertain, its synthesis has not been reported yet [85]. Aspechinulins C (**88**), isolated from the fungus *Aspergillus* sp. FS445, exhibited the most potent inhibitory activities against nitric oxide (NO) production in comparison to other Aspechinulins compounds (IC_50_ = 20–90 mM) [86]. 

Barettin (**89**) is a brominated cyclodipeptide isolated from the boreal sponge *Geodia barrette*. Like its reduced analog 8,9-dihydrobarettin (**69**), it exhibited inhibitory activities, such as potent antifouling, antioxidant, and anti-inflammatory activities, and reduced the DC secretion of IL-12p40 and IL-10 (IC_50_ = 21.0 and 11.8 μM, respectively) [78,79,80]. 

Neoechinulin A (**90**), isolated from the marine-derived fungus *Aspergillus* sp., and Variecolorin O (**91**), extracted and characterized from the *Eurotium* sp. SCSIO F452 fungus, exhibited significant radical scavenging activity against DPPH [42]; compound **90** also showed UV-A protecting activity (IC_50_ = 24 μM) [87]. Isoechinulin A (**92**), isolated from the *Eurotium rubrum* MPUC136 fungus, showed inhibitory activity against the influenza A/WSN/33 virus (IC_50_ = 42.7 μM) [81,82]. 

Compound **93** was isolated from *Eurotium cristatum* EN-220 and showed potent lethal activity against brine shrimp and a weak nematicidal effect against *Panagrellus redivivus* (LD_50_ = 110.3 μg/mL) [83]. 

Neoechinulin B (**94**), Neoechinulin C (**95**), Rubrumline D (**96**), and Rubrumline E (**97**), obtained from the *Eurotium rubrum* fungus, had weak antiviral effects against the influenza virus strain A/WSN/33 (H_1_N_1_) that was propagated in MDCK cells [81]. 

Eurotiumin C (**98**), isolated and characterized from the *Eurotium* sp. SCSIO F452 fungus, showed significant radical scavenging activities against DPPH (IC_50_ = 13 μM) [42]. 

Photopiperazines A–D (**99**–**102**), unsaturated diketopiperazine derivatives, were isolated from the *Actinomycete bacterium* strain AJS-327 and exhibited selective toxicity toward U87 and SKOV3 lines (IC_50_ = 4.1 × 10^−4^ μM and 7.5 × 10^−4^ μM) [88] (Figure 24). 

##### Bis-Indole Diketopiperazine

In this subsection, naturally occurring DKPs with biological activity that contain two indole units are been summarized (Figure 25) [89]. Aspergilazine A (**103**), isolated from the marine-derived fungus *Aspergillus taichungensis* ZHN-7-07, contains a rare N1 to C6 linkage between two DKPs. Compound **103** has weak activity against the influenza A (H_1_N_1_) virus with an inhibition of 34.1% at a concentration of 50 μg/mL [90,91]. 

Brevianamide S (**104**), extracted from the marine-derived fungus *Aspergillus versicolor*, showed antibacterial activity against BCGs (IC_50_ = 9.0 µM) [73,74]. 

Dinotoamide J (**105**), obtained from a marine-derived fungus called *Aspergillus austroafricanus* Y32-2, demonstrated angiogenesis-promoting activity and exhibited proangiogenic activity in a PTK787-induced vascular injury zebrafish model [92]. 

##### Synthetic Routes of DKPs

To obtain indole DKP derivatives, there are two key steps in every synthetic route: the synthesis of the DKP core, and the coupling of the DKP and the indole unit. Regarding the construction of the DKP ring, three immediate disconnections of a 2,5-diketopiperazine ring can be envisioned: the amide bond at N1–C2 (**A**), the N1–C6 bond (**B**), and the C5−C6 bond (**C**). Additionally, two other possible disconnections involving two bonds can be devised: tandem cyclization via N1−C2/C3−N4 (**D**) and via C2−N1− C6 (**E**) (Figure 26). 

The amide bond formation (A) can be carried out through four different methods: dipeptide formation followed by cyclization, Ugi chemistry, amino acid condensation, and Aza-Wittig cyclization. The *N*-alkylation synthesis (B) can be approached in three different ways: using α-haloacyl derivatives of amino acids, the aza-Michael reaction, and the Diels−Alder reaction. The approach C can occur via C–C cyclization radical-mediated or enolate acylation [71]. The tandem cyclization synthesis (D and E) can be regarded as extensions of (A–C), and they share some common processes in tandem fashion. 

Given the straightforward character of the procedure and the huge chiral pool of commercially available α-amino acids, there are several synthetic examples of the dipeptide route using different coupling reagents. 

A representative case of an intramolecular aza-Wittig reaction forming the 2,5-DKP ring is provided in Figure 27. The acylation between amino acids esters (**106**) and chloroacetyl chloride, followed by treatment with sodium azide (NaN_3_) and Ph_3_P, creates 2,5-diketopiperazine **107** via iminophosphorane intermediates [93]. 

Considering the obtention of indole DKPs, many methods of synthesis and biosynthesis have been described over decades, but surprisingly very few of them were employed to create biologically active compounds. As illustrative examples, the reported synthesis and biosynthesis of Brevianamide F (**64**) are depicted in Figure 28. Nicolás et al. carried out a solid phase methodology following the Ashnagar synthesis, which furnished **63** in very good yields but required the installation–removal of protecting groups (Figure 28a) [94,95]. On the contrary, the biosynthesis approach of **64** uses directly unprotected L-Trp (**20**) and L-Pro (**108**) as precursors. This way, using FtmPS (a nonribosomal peptide synthetase) from *Aspergillus fumigatus* as a catalyst, Brevianamide F (**64**) can be obtained (Figure 28b) [69]. 

The synthesis of Neochenulin A (**94**) is an example of a diketopiperazine attached at C3 with an ethylidene bridge. The reaction of the aldehyde **109** and diketopiperazine **110** promoted *t*-BuOK in DMF and created the C3-ethylidene-bridged indole DKP core in one step. The subsequent deacetylation and elimination of the methoxymethyl group (MOM) created target compound **94** (Figure 29) [96]. 

An example of the synthesis of a dimer of the natural product brevianamide F (**64**), aspergilazine A, involves a selective palladium-catalyzed indole N-arylation with brevianamide F (**64**) and *N*-Boc bromo derivative **111**. It had an excellent yield of the product **112**, which, upon facile deprotection, formed Aspergilazine A (**103**) (Figure 30) [75]. 

##### DKPs Featuring Dimethylpyranoindole

Firstly, DKPs containing a hydropyran[3,2-*f*]indole nucleus were described. In this sense, Asperversamides (**113**–**116**) (Figure 31) were extracted from the filamentous fungus *Aspergillus versicolor*, collected from the mud in the South China Sea [97]. All of them contain a rare, linearly-fused dimethylpyranoindole. All these DKP alkaloids exhibited potential iNOS inhibitory activities, related to anti-inflammatory activity. The best IC_50_ value was for compound **114** (5.39 μM), whose planarity was found to be important for its binding capacity to form strong hydrogen bonds with the HEME [97]. Studies of structural elucidation showed that compound **113** is a C17 epimer of Dihydrocarneamide A (**117**). This carneamide derivative, and Iso-notoamide B (**118**), came from the marine-derived endophytic fungus *Paecilomyces variotii* EN-291 and exhibited weak cytotoxic activity against NCI-H460 (IC_50_ = 69.3 and 55.9 μM, respectively) [98]. 

Notoamides are a large family containing a hydropyran[3,2-*e*]indole, isolated from the *Aspergillus* species of fungi. Biosynthetically, they are related to breviamides, paraherquamides, marcfortines, sclerotiamides, asperalines, avrainvillamides, and stephacidins [99,100]. The presence of a bicyclo[2.2.2]diazaoctane (Figure 32) in their structures causes many of these alkaloids to display a variety of biological activities [101]. Thus, Notoamides (**119**–**122**) showed moderate cytotoxicity against HeLa and L1210 cell lines (IC_50_ = 22–52 μM). Furthermore, Notoamide C (**121**) and 5-Chlorosclerotiamide (**123**) had potent anti-fouling and antilarval settlement activities against *Bugula neritina* [102]. Likewise, 17-O-ethylnotoamide M (**124**) did not display cytotoxicity against non-malignant HEK 293 T9 and MRC-9 cell lines and inhibited the colony formation of 22Rv1 cells, related to resistance against hormone therapy for prostate cancer [103]. 

6-*epi*-Avrainvillamide (**125**) and 6-*epi*-Stephacidin A (**126**) were isolated from *Aspergillus taichungensis* and exhibited significant activities against HL-60 (IC_50_ = 4.45 and 1.88) and A549 (3.02 and 1.92) cell lines [104]. Asperthins A,F (**127**,**128**), extracted from a culture of *Aspergillus* sp. YJ191021, displayed moderate anti-inflammatory activity by measuring the secretion of the inflammatory factor 1L-1β by THP-1 cells [105]. Versicamide H (**129**), containing an eight-membered hexahydroazocine ring, was obtained from *A. versicolor* HDN08-60 and showed moderate activity against HeLa, HCT-116, HL-60, and K-562 cell lines and PTK inhibitory activities [106]. 

##### Synthesis of Brevianamides Bicyclo[2.2.2]diazaoctano Alkaloids

The synthetic approach to brevianamides, from 1998 to 2017, has been reviewed by Lawrence et al. [107]. Recently these authors have developed a unified biomimetic synthetic strategy for preparing many of the known bicyclo[2.2.2]diazaoctane brevianamides (Figure 33). 

The synthesis starts with the preparation of (+)-Dehydro-deoxybreviamide E (**130**) from L-tryptophan (**20**), in a five-step gram-scale procedure. Subsequent treatment with *m*CPBA, followed by exposure of the obtained dehydrobrevianamides E (**131**) to LiOH/H_2_O in water at room temperature, created the natural (+) enantiomers of Breviamide A (**132**) and B (**133**). The treatment of **130** with NCS, and then LiOH/H_2_O, produced Brevianamide X (**134**) and Z (**135**).

##### Synthesis General of Hydropyranoindole Alkaloids

The synthesis of natural products bearing a pyranoindole nucleus has been reviewed by Catalano et al. [108]. As seen, some marine indole alkaloids have a hydropyrano ring fused to the pyrrole in a linear or angular manner. In Figure 34, the last step of both synthetic procedures is shown [109,110].

##### Spirocyclic DKP Alkaloids

These prenylated indoles contain a spirocycle in their structures, linked at the indole or at diketopiperazine rings (Figure 35). 

Eurotinoids A–C (**136**–**138**) were characterized from the sediment-derived fungus *Eurotium* dp. SCSIO F452. All the spirocyclic alkaloids showed significant radical scavenging activities against DPPH (IC_50_ = 3.7–24.9 μM) [111].

Spirotryprostatin E (**139**) was isolated from the holothurian-derived fungus *Aspergillus fumigatus* and showed cytotoxicity against MOLT-4, A549, HL-60, and BEL-7420 [112].

Dihydrocriptoechinulin D (**140**) was isolated from a mangrove-derived fungus, *Aspergillus effuses* H1-1, and showed activity against P388 and HL-60 cell lines and inhibitory activity against topoisomerase I [113].

Variecolorins A–C (**141**–**143**) were characterized by the sediment-derived fungus *Eurotium* sp. SCSIO F452. (+)-**141** exhibited stronger antioxidative activity than (−)-**141** against DPPH (IC_50_ = 58.4 μM and 159.2 μM respectively), while (+)-**142** and (+)-**143** showed more potent cytotoxicity against SF-268 (IC_50_ = 12.5 and 30.1 μM) and HepG2 cell lines (IC_50_ = 15.0 and 37.3 μM). (−)-**142** and (−)-**143** were inactive (IC_50_ > 100 μM), which indicated that different enantiomers might result in different biological activities [114].

Variecolortides A–C (**144**–**146**) were obtained from a halotolerant fungus, *Aspergillus variocolor* B17, and displayed weak cytotoxicity towards the K562 human leukemia cell line [19]. They also showed an interesting caspase-3 inhibitory activity (associated with cellular apoptosis) [115].

##### Other Polycyclic DKP Alkaloids

These prenylated indoles contain a variable number of cycles in their structures. They are presented below in increasing order of complexity (Figure 36).

Two Fumitremorgin B (**147**,**148**) derivatives were isolated from the holothurian-derived fungus *Aspergillus fumigatus* and showed similar bioactivity to Spirotryprostatin E, previously described [112]. A structural analog, 13-*O*-Prenylverruculogen (**149**), containing a dioxolane cycle, exhibited potent insecticidal activity against brine shrimp (artemia salina) [116]. On the other hand, Prenylcycloprostratin (**150**) and 9-Hydroxifumitremorgin C (**151**), obtained from *A. fumigatus* YK-7, displayed activities towards U937 cell lines [117].

Drimentine G (**152**), isolated from marine-sediment actinomycete *Streptomyces p.* CHQ-64, showed cytotoxic activities against HCT-8, Bel-7402, A549, and A2780 cell lines [118].

Brevicompanins (**153**–**158**) were isolated from the fungus *Penicillium brevicompactum* and exhibited anti-inflammatory activity associated with BV2 microglial cell lines [119]. Compound **153** also showed antiplasmodial activity. A structural analog, Shornephine A (**159**), with a diketomorpholine ring, was isolated from the marine sediment-derived *Aspergillus* sp. (CMB-M081F) and was identified as a non-cytotoxic inhibitor of the P-glycoprotein associated with MDR cancer cells [120].

Okaramine S (**160**) was produced by Aspergillus taichungensis ZHN-7-07, isolated from the rhizosphere soil of the mangrove plant Acrostichum aureum. It exhibited cytotoxic activity against HL-60 and K562 cell lines with IC_50_ values of 0.78 and 22.4 μM, respectively [121].

Deoxyisoaustamide derivatives (**161**,**162**), containing an eight-membered hexahydroazocine ring, were extracted from the fungus *Penicillium dimorphosphorum* KMM 4689 from soft coral samples. These compounds showed neuroprotective activity against the acute toxicity of paraquat (PQ) murine neuroblastoma Neuro-2a cells [103], with no cytotoxicity towards these neuro-cells.

Raistrickindole A (**163**), containing an oxindole ring, was extracted from *Penicillium raistrickii* IMB17-034 and showed activity against the hepatitis C virus (HCV) with an EC_50_ value of 5.7 μM [122].

Indotertine B (**164**) was isolated from the marine sediment-derived actinomycete *Streptomyces* sp. CHQ-64 [123] and exhibited cytotoxic activities against HCT-8, Bel-7402, A549, and A2780 cell lines with IC_50_ values of 2.81, 1.38, 1.01, and 2.54 μM, respectively [124]. 

Nocardioazine A (**165**), isolated from a marine sediment-derived bacterium, *Nocardiopsis* sp. (CMB-M0232) is an effective and noncytotoxic inhibitor of the multidrug resistance factor P-glycoprotein and is able to reverse resistance in SW620 Ad300 cells [125].

##### General Synthesis of Indole DKP Alkaloids

A general strategy for the synthesis of indole DKP alkaloids (Figure 37) has been described by Jia et al. [126]. Three types of analogs of indole DKP alkaloids were synthesized: fused pentacyclic indole DKPs (**166**), trypostatin open-ring indole DKPs (**167**), and spiropentacyclic indol DKPs (**168** and **169**).

The Pictet–Spengler reaction of methyl L-tryptophan hydrochloride **170** with several aldehydes leads to the corresponding chiral cyclic intermediate **171**. The subsequent reaction of **171** with F-moc-L-Pro-Cl created **172**, which, by treatment with morpholine, produced the fused pentacyclic indole DKP (**166**). When compound **172** is treated with NBS, it undergoes a spiro rearrangement providing the corresponding spiro-pentacyclic indoles, which, upon treatment with morpholine, generates the DKP derivative, **168** (R = alkyl). When the substituents are aromatic, open-ring indoles (**167**) are formed. Another approach for the preparation of the spiro-pentacyclic scaffold (**169**, R = aryl) used a 1,3 dipolar cycloaddition of 2-oxoindolin-3-ylidenes with azomethine ylides, followed by the previously described procedure (treatment with F-moc-L-Pro-Cl and morpholine).

#### 2.2.2. Hexahydropyrrolo[2,3-b]indol (HPI) Derivatives

In this kind of alkaloid, the indole group from tryptophan is fused with an additional pyrrole ring (Figure 38), highlighted by a group of Flustramines isolated from the marine bryozoan *Flustra foliacea* [127]. The simple Flustramine C (**173**) showed activity to inhibit biofilm formation in *A. baumannii*, a human pathogen associated with hospital-acquired infections. A structural modification by adding a triazole amide moiety with a large hydrophobic chain at pyrrroloindole (**174**) increased the antibiofilm activity, from IC_50_ values of 174 μM to 3.4 μM, respectively [117]. 

##### Synthesis of Hexahydropyrrolo[2,3-b]indol (HPI) Derivatives

Several procedures have been described for the synthesis of a pyrroloindole scaffold. Below, the focus is on the synthetic routes for the preparation of Flustramines (Figure 39 and Figure 40) and on the known routes to build the HPI tricycle skeleton (Figure 41). 

##### Synthesis of Flustramines

The general approach to Flustramines consists of tandem olefination, isomerization, and Claisen rearrangement to provide the intermediate **175**. The successive deacetylation, and selective reduction of the nitrile group of compound **176** with subsequent cyclization, leads to pyrroloindole **177**. A final methylation step creates Flustramine C (**178**) **[35]**.

Bunders et al. [117] described an effective method to obtain Flustramine analogs **179** with a general scaffold. As indicated in Figure 40, a Fischer indolization reaction of hemiaminal **180** created the tricyclic core **181**. The corresponding functionalization of **181** and final deprotection created the aforementioned product **179**. 

##### Synthesis of HPI Tricyclic Skeleton

The synthesis of HPIs has been quite extensively reviewed by Albericio et al. [128]. Figure 41 shows the most significant synthetic routes to obtain a wide variety of HPI alkaloid derivatives, using functionalized indoles, oxidized indoles, and tryptamines as starting materials. The usually described procedures involve classic approaches by cyclization, including acid-catalyzed, oxidative, reductive, and alkylative, with nucleophiles. Other procedures take place by [3,3]-sigmatropic rearrangement and Fischer indolization. On the other hand, complex structures were obtained by modern procedures, including Pd-catalyzed reactions such as Larock heteroannulations or aza-Pauson–Khand cyclocarbonylation.

#### 2.2.3. Indolactam Alkaloids

Teleocidin analogs **182** and **183** were isolated from different *Streptomyces* sp., obtained from marine sponges. The first compound, **182**, had neurological activity via the protein kinase C (PKC) pathway [37], while the second compound, **183**, exhibited cytotoxicity against HeLa and ACC-MESO-1 cell lines (Figure 42).

Pendolmycin analogs **184** and **185** were isolated from actinomycete *Marinactinospora thermotolerans* SCSIO 00652. They showed antiplasmodial activities against the *Plasmodium falciparum* strains 3D7 and Dd2 [129].

#### 2.2.4. Other Polycyclic Indole Alkaloids

Pentacyclic carbazole derivatives Xiamicyn A (**186**) and B (**187**) were isolated from different endophytic *Streptomyces* sp. Compound **186** was an anti-HIV agent [39], while compound **187** exhibited potent antibacterial properties (Figure 43) [122].

Fusaindoterpenes A (**188**) and B (**189**), extracted from a culture of *Fusarium* sp. L1, showed interesting antiviral activity against the Zika virus with EC_50_ values of 12 and 7.5 μM, respectively. The structure–activity relationship study of these compounds revealed that the cyclopentane-pyrrole fused ring is essential for higher antiviral activity [130]. 

Penerpenes A–B (**190**,**191**) are two indole diterpenoids obtained from *Penicillium* sp. KFD28, isolated from a bivalve mollusk. Both compounds displayed inhibitory activities against PTPs, becoming a promising target for drug discovery against diabetes [131,132]. 

Shearinines D and E (**192**,**193**) were isolated from the marine-derived strain of the fungus *Penicillium janthinellulm* Biourge [131]. Both compounds exhibited varied bioactivity, such as the induction of apoptosis in the human leukemia cell line HL-60 [131], as well as inhibition against *Candida albicans* biofilm formation [132]. 

Spirocyclic Citrinadin B (**194**) was extracted from *Penicillium citrinum*, obtained from a red alga, and showed cytotoxic activity against murine leukemia L1210 cells [133].

Triaza-spirocyclic Meleagrins B–E (**195**–**198**) were isolated from the fungus *Penicillium* sp. and showed cytotoxicity against HL-60, MOLT-4, A549, and Bel-7402 cell lines. The bioactivity increases with the complexity of the Meleagrins, being lower for D and E than for B and C [134,135].

Penitrem derivatives (**199**–**201**) were isolated from the marine-derived fungus *Penicillium* commune and *Aspergillus nidulans* EN-330. Compound **199** showed significant anti-invasive and antiproliferative activity against MCF-7 and MDA-MB-231 tumor cell lines [136]. The other two Penitrems exhibited antimicrobial activity [137].

Asperindoles A (**202**) and Ascandinine D (**203**) are indolediterpenes with the same structural scaffold obtained from the culture of two different *Aspergillus* sp. Compound **202** exhibited toxicity against 22Rv1 (induction of cellular apoptosis), PC-3, and LnCaP prostate cancer cell lines [138], while **203** was active against the HL-60 (promyelocytic leukemia) cell lines [139]. 

#### 2.2.5. Ergot Alkaloids

Pibocins A and B (**204**–**205**) and Fumigaclavine A (**206**) are examples of Ergot alkaloids with interesting bioactivity (Figure 44). Pibocins were isolated from ascidian *Eudistoma* sp. [140] and were found to have antimicrobial and cytotoxic effects against mouse Ehrlich carcinoma cells [140,141]. Compound **206** was extracted from the fungus *Aspergillus fumigatus* [142] and induced apoptosis in MCF-7 breast cancer cells [143].

### 2.3. Annelated Indole Alkaloids

Within this subsection, alkaloids containing a single indole core fused with no prenyl-derived (hetero)cyclic ring systems are disclosed (Figure 45).

#### 2.3.1. Quinazoline(inone)-Containing Annelated Indole

Aspertoryadins F and G (**207**–**208**) contain a 2-indolone moiety linked to a quinazolinone ring through a five-membered spiro lactone. Both compounds were extracted from *Aspergillus* sp. from a bivalve mollusk. They exhibited quorum sensing (QS) inhibitory activity against *Chromobacterium violaceum* CV026, causing skin infections. These compounds prohibited bacterial pathogenicity [142].

Fumigatoside E (**209**) was obtained from *Aspergillus fumigatus* SCSIO 41012 and showed moderate to strong antibacterial and antifungal activity, with LC_50_ values of 6.25 μM, against *A. baumannii* 15,122 and *S. aureus* ATCC 16,339, and 12.5 μM against *A. Baumannii* ATCC 19,606 and *K. pneumoniae* ATCC 14,578. Strong activity against *F. oxyosporum f.* sp. (LC_50_ = 1.56 μM) was also observed [144].

Fumiquinazoline J (**210**) was isolated from the fungal strain *Aspergillus fumigatus* H1-04 and exhibited cytotoxicity against the cell lines ts FT210, P388, HL-60, A549, and Bel-7402 [143].

Cottoquinazoline D (**211**), obtained from the marine-derived fungus *Aspergillus versicolor*, was reported to show antifungal activity against *C. albicans* [145,146].

Scequinadoline A (**212**) and Scedapin C (**213**) contain an imidazoindolone ring and were isolated from an extract of the soft coral-associated fungus *S. apiospermum* F41-1. Both compounds displayed significant anti-HCV activity against the J8CC recombinant [147].

#### 2.3.2. Imidazolone-Containing Pyrrolidinone

Securamines H and I (**214**–**216**) are hexacyclic annelated indole alkaloids isolated from the bryozoan *Securiflustra securifrons* that showed potent cytotoxicity against A2058, HT-29, and MCF-7 lines (Figure 46) [148].

#### 2.3.3. β-Carbolines

β-Carboline alkaloids (βCs) are a tryptophan-derived family of natural products whose basic structure derives from the tricyclic 9*H*-pyrido[3,4-*b*]indole (Figure 47). Although initially discovered in plants, a wide range of these compounds have been isolated over decades from marine sources, such as tunicates [149], sponges [150], and bryozoans [151]. βCs display a wide range of outstanding biological activities and, to the best of our knowledge, several plant-isolated and synthetic representative examples, depicted in Figure 47, have been approved by the FDA and commercialized as drugs at some point, including Taladafil [152] and Yohimbine [153] for treating erectile dysfunction, Reserpine [154], Deserpidine [155] and Rescinnamine [156] for treating hypertension, Abecarnil [157] as an anxiolytic, and Cipargamin [158] as an antimalarial. 

However, no example of a marine-derived βC has been approved by the FDA, to the best of our knowledge. This is quite surprising since, as will be showcased in the next subsections, they can exert a wide variety of biological activities, such as anticancer, antibiotic, antiplasmodial, anti-inflammatory, and antifungal, among others. 

βCs can be found in nature in a monomeric or dimeric fashion [159]. However, some of them are hybrid structures with two different βC cores. Therefore, monomers and dimers will be disclosed in separate subsections and, attending to the absence or presence of extra fused rings in the basic βC skeleton, monomers will be subsequently grouped as ‘simple’- and annelated-βCs.

##### β-Carboline Monomers


‘Simple’ β-Carbolines


Regarding the saturation of the indole-fused pyridine ring, these compounds can be classified as β-carbolines (βCs), dihydro-β-carbolines (DHβCs), and tetrahydro-β-carbolines (THβCs). It is worth mentioning that the *N*-methyl quaternary salt of β-carboline alkaloids also occurs in nature. 

The simplest β-carboline, Norharmane (**217**), first isolated from a higher plant, can be found in different marine sponges (Figure 48). In 2007, Herraiz et al. showed that **217** has possible applications against PD [160]. 

The presence of substituents in the basic structure of βC, and the level of reduction of the ring, lead to enhanced or new properties in comparison with **217**. The rest of the section has been structured according to the substituted position in the βC, which is responsible for the therapeutic activity, trying to group them in their corresponding families and making a comparison with their reduced analogs when possible. Therefore, the following subsections will be presented: C1-substituted-βCs, Manzamines, N2-substituted-βCs, and C3-substituted-βCs. It is important to remark that, although manzamines belong to C1-substituted-βCs, their specific structure and bioactivities require a separate discussion from their simpler analogs.

##### C1-Substituted (DH/TH)β-Carbolines

βCs in which the C1-substitution is responsible for their therapeutic activity represent the largest family of these scaffolds. The variety of functional groups that can be found at C1 is pretty wide, ranging from simple alkyl chains or aryl groups to complex glycosides or polycycles. 

Harmane (**218**) was isolated from the culture of the marine-sponge-associated fungus *Neosartorya tsunodae* KUFC 9213 [161]. Compound **218** exhibited stronger AChE and BuChE inhibition (IC_50_ > 10 μM) compared to **217** and weak in vitro antileishmanial activity against *Leishmani infantum* [162]. 1-Ethyl-β-carboline (**219**), isolated from the marine bryozoan *Orthoscuticella ventricosa*, exhibited moderate antiplasmodial activity (IC_50_ = 18 μM) against the *P. falciparum* K1 strain [151]. The addition of a C4-OMe to the pyridine ring (**220**) exerted a detrimental effect on the activity [163]. Other βCs from the same bryozoan, such as 1-ethyl-4-methylsulfone-β-carboline (**222**), Orthoscuticelline C (**223**), and Orthoscuticelline D (**224**), had lower efficiency, indicating that the addition of C4–sulfone to the ring, or hydroxy, amino, or sulfonic acid groups to the alkyl chain, were not beneficial [150,164] (Figure 49). 

However, Harmine (**221**), a C7-OMe analog of **217**, first isolated from plants but widely found in marine species, exhibited a wide range of bioactivities, including antitumor, antibiotic, antifungal, antioxidant, antiplasmodial, antimutagenic, and antigenotoxic activity. Further, it acts on gamma-aminobutyric acid type A and the monoamine oxidase A or B receptor, improves insulin sensitivity, exerts vasorelaxant effect, and suppresses osteoclastogenesis, among others. These properties have been well documented by Patel and coworkers [164].

Eudistalbin A (**225**), isolated from a tunicate *Eudistoma album*, presented in vitro cytotoxicity (IC_50_ = 3.2 μg/mL) against KB cells [149]. Plakortamine A (**226**), isolated from the sponge *Plakortis nigra*, showed antitumor activity against the HCT-116 cell line (IC_50_ = 3.2 μM) [150]. Both Eudistomidin C (**227**) and J (**228**), obtained from tunicate *Eudistoma glaucus* [165], have potent cytotoxicity against murine leukemia L1210 cells (IC_50_ = 0.36 and 0.047 μg/mL, respectively) [165,166], while only **228** is active against P388 and KB cancer cells (IC_50_ = 0.043 and 0.063 μg/mL, respectively) [166]. 14-Methyleudistomidin C (**229**), from the ascidian *Eudistoma gilboverde*, demonstrated significant cytotoxicity against four different human tumor cell lines (IC_50_ < 1.0 μg/mL) [167]. Ingenine E (**230**), isolated from the sponge *Acanthostrongylophora ingens*, is strongly cytotoxic against MCF-7, HCT-116, and A549 lines [168]. It is worth mentioning that, although Orthoscuticelline C (**222**) is chemically similar to **215**–**228**, its anticancer biological activity has not been tested so far.

Opacalines A (**231**) and B (**232**), found in the ascidian *Pseudodistoma opacum*, exhibited antiplasmodial activity due to alkyl guanidine-substituted chains (IC_50_ = 2.5 and 4.5 μM, respectively) [169]. As observed, the N9-hydroxylation reacts negatively to this activity. Other synthetic debromo- or THβCs derivatives of **231** and **232** were less active than the parent compounds, indicating that the Br atom plays an important role in the activity.

Eudistomins W (**233**) and X (**234**), isolated from tunicate *Eudistoma* sp., have antifungal activity against *C. albicans* and *B. subtilis*, *S. aureus*, and *E. coli*, respectively, as well as some antibiotic properties [170].

Imidazolium-containing Gesashidine A (**235**), first isolated from a *Thorectidae* sponge, showed antibacterial activity against Micrococcus luteus but no cytotoxicity against the cell line L5178Y [171]. Interestingly, the presence of a C3-carboxylate shuts down the antibacterial activity of Dragmacidonamine A (**236**), isolated from the same sponge, and its sulfoxide Hyrtimomine H (**237**), obtained from *Hyrtios* sponge. However, it enhances their cytotoxicity when compared to **235** (Figure 49). 

Reduced DHβC and THβC analogs of compounds **217**–**237** (Figure 50) have similar therapeutic activity compared to their unsaturated counterparts. Eudistomidins B (**238**), G (**239**), H (**240**), and I (**241**), isolated from *Eudistoma glaucus*, exhibited cytotoxicity against L1210, L5178Y, P388, and KB cancer cells, although weaker than related compounds **223**–**237**. Ingenine F (**242**), obtained from *Acanthostrongylophora ingens*, showed similar levels of cytotoxic activity against MCF-7, HCT-116, and A549 lines compared to compound **230** [172]. (+)-7-Bromotrypargine (**243**), isolated from the marine sponge *Ancorina*, exerts antimalarial activity similar to **231**, but also weak cytotoxicity against HEK293 cells [173]. Haploscleridamine (**244**), isolated from *Haplisclerida* sponge, was identified as an inhibitor of cathepsin K [174], while its C3-CO_2_H analog Hainanerectamine C (**245**), identified from the *Hyrtios erecta* sponge, showed moderate anticancer activity as an inhibitor of Aurora kinase A [35].

Hyrtimomine I (**246**) and **J** (**247**), hydroxyimidazolium βCs found in the *Hyrtios* sponge, exhibited antifungal activity against *A. niger* (IC_50_ = 8.0 μg/mL each) and *C. albicans* (IC_50_ = 2.0 μg/mL each), but only **246** showed activity against *C. neoform* (IC_50_ 4.0 μg/mL). However, Hyrtimomine H (**248**), from the same sponge, showed no activity, indicating that the C3-CO_2_H group is crucial [175] (Figure 51). It is worth noting that this kind of activity has not been reported so far for similar compounds **235**–**237**. 

Blunt and Munro indicated that C1-vinyl groups might be beneficial for antitumor activity (Figure 52). 1-Vinyl-8-hydroxy-β-carboline (**249**), collected from bryozoan *Cribricellina cribaria* [176], and Plakortamine B (**250**), produced by the sponge *Plakortis nigra* [150], were found to be active against the P388 (IC_50_ = 100 ng/mL) and HCT-116 cell lines (IC_50_ = 3.2 μM), respectively. The C1-aryl compound Chaetogline F (**251**), obtained from the fish-derived fungus *Chaetomium globosum* 1C51 through biotransformation [177], represents a more promising structure for the design of anti-Alzheimer’s drugs [178] and had antibiotic activity against *Veillonella parvula*, *Bacteroides vulgatus*, *Streptococcus* sp., and *Pepto streptococcus* sp. [179]. Apart from antibiotic activities, other authors found that some synthetic C1-aryl derivatives exhibited activity against *Leishmania donovani* [180].

C1-furyl-substituted Flazin (**252**) (Figure 53), obtained from the oyster *Crassostrea sikamea* [181], is a promising candidate for the development of anti-HIV drugs [182]. An exhaustive SAR study carried out by Liu et al. identified the synthetic Flazinamide (**253**) as the most promising drug. Eudistomin I (**254**), isolated from *Eudistoma olivaceum* tunicate, contains a dihydropyrrole ring that confers its antibacterial effects [183,184,185]. Indole-substituted Eudistomin U (**255**) and Isoeudistomin U (**256**), isolated from *Lissoclinum fragile*, and their synthetic analogs, have been reported to have antibacterial, antimalarial, and anticancer properties, as extensively reviewed by Kolodina and Serdyuk [186]. Plakortamine D (**257**), a C1-isoxazolidine-substituted scaffold obtained from the *Plakortis nigra* sponge, has antitumor activity against the HCT-116 cell line (IC_50_ = 15 μM) [150]. Finally, Annomontine (**258**), Ingenine C (**259**), and Ingenine D (**260**), all of them bearing aminopyrimidine rings and isolated from the Indonesian sponge *Acanthostrongylophora ingens*, exhibited cytotoxic activities against MCF-7 and HCT-116 [168,187].

1-Acetyl-β-carboline (**261**), isolated from *Marinactinospora thermotolerans*, showed weak cytotoxicity against NCI-H460 cells (IC_50_ = 18.73 μg/mL) [188] and antibiotic properties against *S. aureus* [189]. Eudistomidin K (**262**), from the tunicate *Eudistoma glaucus*, exhibited weak cytotoxicity against P388, L1210, and KB cells (IC_50_ > 10.0 μg/mL) [166]. Marinacarbolines A–D (**263**–**266**), obtained from *Marinactinospora thermotolerant*, and their synthetical derivatives, bear an additional C3-amido moiety with pendant aryl rings (Figure 54). Their cytotoxicity was first investigated in 2015 [190]*,* but Hong and Lee have performed a very recent and in-depth SAR study against ocetaxel-Resistant Triple-Negative Breast Cancer [191]. Compounds **263**–**266** also exhibit promising antimalarial activity [129]. Eudistalbin A (**267**), isolated from *Eudistoma album* tunicate, exerts cytotoxic activity in vitro against KB cells (IC_50_ = 3.2 μg/mL) [149]. Eudistomin T (**268**), from the tunicate *Eudistoma olivaceum*, exhibited not only weak phototoxicity but also antibiotic properties [184].

Eudistomin Y (**269**), isolated from *Eudistoma* tunicates, and its synthetic analogs, tends to exhibit antifungal [192] and antibiotic [192,193] properties (Figure 55)*,* but also significant cytotoxic and antiproliferative activities [192,194,195]. SAR analysis indicated that an increased number of Br atoms in the aromatic rings increased their antibiotic effect. Reduction of the benzoyl moiety does not affect its properties, as found for Eudistomin Y_11_ (**270**).

Xestomanzamine A (**271**) (Figure 56), isolated from the sponge *Acanthostrongylophora* sp., had moderate antibiotic, anti-HIV, and antifungal activity, but no cytotoxicity against A594 and HCT-116 [196]. However, imidazole-containing Hyrtiocarboline (**272**), from *Hyrtios reticulatus* sponge, showed significant cytotoxicity against H522-T1, MDA-MB-435, and U937 cell lines (IC_50_ = 1.2, 3.0, and 1.5 μg/mL, respectively) [197]. Imidazolium-containing Hyrtiomanzamine (**273**), from *Hyrtios erecta* sponge, and Dragmacidonamine A (**274**), from *Dragmacidon* sponge, exhibited some cytotoxicity [171,197]. Further, **273** exhibited some immunosuppressive activity [198]. Indolyl-substituted Pityriacitrin (**275**), first isolated from a *Paracoccus* marine bacterium, exerts promising anticancer activity against MCF-7, MDA-231, and PC3 cell lines [199]. In-depth SAR analysis of Pityriacitrin analogs showed that C3 amide, hydrazide, hydrazones, 1,3,4-oxadiazole, 1,2,4-triazole, and pyrazole moieties are essential for potent anticancer activity [200].

Hyrtiosulawesine (**276**), found in the Indonesian sponge *Hyrtios erectus*, displays a great variety of properties, such as antioxidant [201], antiphospholipase A2 [202], antidiabetic [203], anti-inflammatory [204], antimalarial [205], and cytotoxicity properties, towards the Hep-G2 cell line (IC_50_ = 19.3 μmol/L) [206]. 6-O-(β-glucopyranosyl)hyrtiosulawesine (**277**), from the same marine species, is only slightly cytotoxic towards hepatic cells and has antimalarial activity (IC_50_ = 5 μM).

Finally, Shishijimicin A–C (**278**–**280**) (Figure 57), isolated from sea squirt *Didemnum proliferum*, has antitumor activity against P388 cells [207]. This property is attributed to the intricate and conjugated enediyne functional group, with **278** being the most powerful enediyne-based antitumor and antibiotic identified to date. Remarkably, the total synthesis of compound **278** was accomplished in 2015 by Nicolaou [208].

##### Manzamines

Manzamines are a special family of C1-substituted βCs in which the C1 moiety generally consists of a characteristic complex penta-or tetracyclic system or a monomacrocycle (Figure 58). Manzamine A (**281**) (also named Keramamine A) [209] was the first reported member of these compounds [210]. Compound **281** showed a broad spectrum of biological effects, including potent antileishmanial and antimycobacterial activity [211], cytotoxicity against pancreatic cancer, P388, and human colorectal carcinoma [210,212,213], and anti-Alzheimer’s activity [214]. It also exhibited antiviral effects against HSV-1, HSV-2, and HIV [211,215,216]. Compound **281** exhibited potent antitubercular activity against *M. tuberculosis* (H37Rv) [217]. 8-Hydroxymanzamine A (**282**) (also named manzamine G or manzamine K) exhibited moderate antitumor activity against KB and LoVo lines and anti-HSV-2 activity [216]. *ent*-8-Hydroxymanzamine A (**283**) is active against P388 (IC_50_ = 0.25 µg/mL) and exerts an in vitro antitrypanosomal effect [218]. Manzamine M (**284**) had cytotoxicity against L1210 cells (IC_50_ = 0.3 µg/mL), and antibacterial activity against *Sarcina lutea* (MIC = 2.3 µg/mL) and *Corynebacterium xerosis* (MIC = 5.7 µg/mL) [219]. 

12,34-Oxamanzamine A (**285**), with a C12–C34 ether bridge, exhibited lower antimalarial and antituberculosis activity compared to the other manzamines [220]. 12,28-Oxamanzamine A (**286**) and 12,28-Oxa-8-hydroxymanzamine A (**287**), with C12–C28 or C12–C34 ether bridges, showed effective antifungal, anti-inflammatory and anti-HIV-1 activities [221]. 

3,4-Dihydro-6-hydroxymanzamine A (**288**) had cytotoxicity against L1210 cells (IC_50_ = 1.4 µg/mL), and antibacterial activity against *Sarcina lutea* (MIC = 6.3 µg/mL) and *Corynebacterium xerosis* (MIC = 3.1 µg/mL) [219]. *N*-Methyl-*epi*-manzamine D (**289**) and *epi*-Manzamine D (**290**) showed cytotoxicity against HeLa and B16-F10 cells [220]. 1,2,3,4-Tetrahydro-2-*N*-methyl-8-hydroxymanzamine A (**291**) (8-Hydroxy-2-*N*-methylmanzamine D) is cytotoxic toward the P388 cell line (ED_50_ = 0.8 µg/mL) [222]. 

Biologically active pentacyclic manzamines having a ketone or alcohol group in their eight-membered ring instead of a double bond, have been also reported (Figure 59). Manzamine E (**292**) and Manzamine F (Keramamine B) (**293**) displayed cytotoxicity toward L5178Y and P388 cells [223]. *Ent*-manzanine F (**294**) inhibited H37Rv (IC_50_ < 12.5 µg/mL) [218]. *ent*-12,34-oxamanzamines E (**295**) and F (**296**) showed weak inhibitory activity against *M. tuberculosis* (IC_50_ value of 128 µg/mL) [220]. Pre-*neo*-kauluamine (**297**) exhibited proteasome inhibitory activity, a potent antitrypanosomal effect, and antimalarial activity [224,225]. 

Several biologically active manzamines containing a βC ring system with a C1-tetracyclic scaffold have been reported (Figure 60). Manzamine J (**298**) showed cytotoxic activity against KB cells (IC_50_ > 10 µg/mL), while its *N*-oxide (**299**) showed cytotoxicity against L1578Y (IC_50_ = 1.6 µg/mL). Additionally, **298** has anti-tubercular activity against H37Rv [217]. Manzamine B *N*-oxide (**300**) displayed weak activity against several Gram-positive and Gram-negative bacteria [226]. Acanthomanzamines D (**301**) and E (**302**), had a strong proteasome inhibitory effect (IC_50_ = 0.63 and 1.5 µg/mL, respectively) [227].

Manzamines H (**303**) and L (**304**) hold cytotoxicity against KB cells (IC_50_ = 4.6 and 3.5, respectively). Compound **304** also possesses weak activity antibiotic activity [226]. Ma’eganedin A (**305**), proved to be a potent antibiotic against *Sarcina lutea* and *B. subtilis* (MIC = 2.8 µg/mL each) [228].

Furthermore, 3,4-Dihydromanzamine J (**306**), and all the aforementioned manzamines, **291, 303**–**305**, showed cytotoxic activity against the L1210 cell line (IC_50_ = 5.0, 2.6, 1.3, 3.7, and 4.4 µg/mL, respectively) [217].

Finally, other types of monomacrocyclics and diverse hexa- and heptacyclic biologically active manzamines have been reported (Figure 61). Manzamine C (**307**) exhibited cytotoxicity against A549, HT-29, and P388 cells (IC_50_ = 3.5, 1.5, and 2.6 μg/mL, respectively) [229]. Pyrrolizine-substituted Kepulauamine A (**308**) unveiled weak inhibition against K562 and A549 cells and moderate antibiotic activity [226]. Manzamine X (**309**) exhibited cytotoxic activity against KB cells (IC_50_ = 7.9 μg/mL) [230], while 6-Deoxymanzamine X (**310**) exhibited cytotoxicity against L5178 cells (ED_50_ = 1.8 µg/mL) [231]. Manadomanzamines A (**311**) and B (**312**) exhibited an anti-tubercular effect (MIC = 1.9 and 1.5 µg/mL, respectively), antiviral activity against HIV-1 (EC_50_ = 7.0 and 16.5 µg/mL, respectively), cytotoxicity against A549 (IC_50_ = 2.5 µg/mL, only **311**) and HCT-116 cells (IC_50_ = 2.5 and 5.0 µg/mL, respectively), and an antifungal effect against *C. albicans* (MIC = 20 µg/mL, only **312**) and *C. neoformans* (MIC = 3.5 µg/mL, only **311**) [196].

##### N2-Substituted (DH/TH)β-Carbolines

The N2-methyl-β-carbolinium salts Irene-carbolines A (**313**) and B (**314**), isolated from ascidian *Cnemidocarpa irene*, exerted anti-Alzheimer’s activity [232] (Figure 62). Notably, other non-brominated derivatives identified in the same species did not exhibit any activity.

The N2-aryl-β-carbolinium species Reticulatol (**315**), Reticulatine (**316**), and Reticulatate (**317**) could be obtained from *Fascaplysinopsis reticulata* sponge. Compounds **316** and **317** had modest antitumor activity, while **315** showed significant selectivity for leukemia [233].

##### C3-Substituted (DH/TH)β-Carbolines

Variabines A (**318**) and B (**319**), with a C3-ester (Figure 63), were isolated from the sponge *Luffariealla variabilis*, and had a respectively little and significant effect on the inhibition of the chymotrypsin-like activity of proteasome and breast cancer metastasis [234]. Therefore, the inhibitory activities are lost by sulfonation of the 6-OH group. Stolonine C (**320**), from the tunicate *Cnemidocarpa stolonifera*, induced apoptosis in the PC3 cell line [235]. Tiruchanduramine (**321**), obtained from the ascidian *Synoicum macroglossum*, could be identified as a promising inhibitor of *α*-glucosidase due to the presence of a cyclic guanidine group [236]. 

C3-indole-substituted βCs have been also found in marine sources, such as the family of Hyrtioerectines isolated from the sponge *Hyrtios erectus*. Hyrtioerectine A (**322**) showed moderate cytotoxicity against HeLa cells (IC_50_ = 10 μg/mL) [237]. Hyrtioerectines D–F (**323**–**325**) exhibited antibacterial behavior against *C. albicans*, *S. aureus,* and *Pseudomonas aeruginosa*. They also exhibited antioxidant activity, and weak antitumor activity against MDA-MB-231, A549, and HT-29 cell lines, with **323** and **324** being more active than compound **325**. Therefore, the methylation of the phenol group hampers the antioxidant activity, while a C4-CO_2_H moiety is more beneficial than an amido group for antitumor properties.

Regarding saturated carbolines (Figure 64), Hyrtioerectine B (**326**) prompted moderate cytotoxicity against HeLa cells (IC_50_ = 5.0 μg/mL).

##### Annelated β-Carbolines

Several βCs with different 5-, 6- or 7-membered fused rings in different positions have been isolated from marine sources over decades, and some of them exhibited promising activities (Figure 65). Fascaplysin (**327**), 3-Bromofascaplysin (**328**), 10-Bromofascaplysin (**329**), 3,10-Dibromofascaplysin (**330**), 6-Oxofascaplysin (**331**), and Homofascaplysate A (**332**) are pentacyclic compounds isolated from the sponge *Fascaplysinopsis* sp., in which the βC core is fused to a 5-membered ring through C1 and N2. In general, Fascaplysin natural and synthetic derivatives represent excellent lead drugs since they exert multiple activities. Namely, anticancer activity against Human Alveolar Rhabdomyosarcoma cells, leukemia, liver cancer cells, melanoma, small lung cancer cells, and ovarian cancer cells, among others. Further, they also exert analgesic, anti-thrombotic, anti-Alzheimer’s, and antimalarial activity [238]. Thorectandramine (**333**), from the marine sponge *Thorectandra* sp., had weak cytotoxicity against MCF-7, OVCAR-3, and A549 cell lines (EC_50_ 27.0–55.0 μg/mL) [239]. 

Eudistomins C (**334**), E (**335**), K (**336**), and L (**337**), Eudistomin K sulfoxide (**338**) and Debromoeudistomin K (**339**) are tetracyclic THβC isolated from different marine ascidians, featuring a fused oxathiazepine ring between C1 and N2, responsible of their antiviral activity against HSV-1 and other DNA- or RNA-viruses [183]. Additionally, **336** has potent antitumor activity against L1210, A549, HCT-8, and P388 cell lines [240].

Hyrtimomines D (340) and E (341), which contain a fused D-ring between C1 and N9 forming a lactam unit, belong to the canthin-6-one family (Figure 66). Both have antifungal activity against *C. albicans* (IC_50_ = 4 and 8 μg/mL, respectively) and *C. neoformans* (IC_50_ = 4 and 8 μg/mL, respectively), but only **340** showed inhibitory activity against *T. mentagrophytes* (IC_50_ = 16 μg/mL) and *S. aureus* (IC_50_ = 4 μg/mL). From their results, the authors inferred that the presence of carboxylic acid is less beneficial for its antifungal properties [175].

##### β-Carboline Dimers

Some recent research has shown a potential trend in which the dimers tend to be more active than the corresponding monomers [159]. Therefore, several authors have turned their attention toward the synthesis and evaluation of these scaffolds. According to the linked positions of the βC monomers, they can be divided into 1,1-, 2,2-, 3,3-, 9,9-linked, and ‘hybrid’ dimers, in which the two βC units are not equivalent. 

However, these structures are not that commonly found in marine species compared to plants and, to the best of our knowledge, only a couple of marine-isolated or marine-inspired synthetic dimers with biological activity have been reported to date.

##### 1,1-Linked Dimers

As far as we can ascertain, only three examples of biologically active marine naturally occurring 1,1-dimers have been reported to date, varying the nature of the organic linker from simple alkyl chains to complex polycyclic structures (Figure 67). Orthoscuticellines A (**342**), a dimer derived from Plakortamine B (**250**) and obtained from the bryozoan *Orthoscuticella ventricosa*, has a 1,2-cyclobutane unit as a linker. Although its *trans* dimer had no activity, **342** demonstrated higher cytotoxicity than parent **250** and moderate antiplasmodial activity [151]. Plakortamine C (**343**), which can be regarded as a Plakortamine A (**226**) dimer and was isolated from the same *Plakortis nigra* sponge, exhibited higher cytotoxic activity than **226** against the HCT-116 cell line (IC_50_ = 2.15 mM) [150]. Finally, the manzamine 1,1-dimer Neo-kauluamine (**344**), isolated from Indonesian *Acanthostrongylophora ingens* sponge, exhibited potent cytotoxic activity against H12999 (IC_50_ = 1.0 mM), proteasome inhibitory activity (IC_50_ = 0.13 mM), and the inhibition of the accumulation of cholesterol esters [224].

It is worth mentioning that, inspired by these structures, Chatwichien et al. developed the synthesis of 1,1-dimers of simple Norharmane (**217**) linked by aminoalkylether chains [241]. Surprisingly, their biological activity against various cancer cell lines was as good as the one exerted by Neo-kualamine (**344**). Given the potential of these compounds, this area is still a hot topic of research with promising expectations. 

##### 9,9-Linked Dimers

Interestingly, an N–N bonded 9,9-dimer of Norharmane (**217**) was isolated from the *Didemnum* sp. ascidian (Figure 68). Although this species’ antibiotic activity was diminished in comparison to **217**, other synthetic derivatives have a wide application. In fact, the double N-methylated carbolinium salt (**345**) was found to be more active for some strains such as *S. aureus* [242].

##### ‘Hybrid’ Dimers

Some manzamine derivatives, in particular in the family of Zamamidines, were found to bear a second pendant βC unit, usually exhibiting an N2–C1′ linkage (Figure 69). Zamamidine C (**346**) demonstrated a potent antitrypanosomal effect against *Trypanosoma brucei brucei* and antimalarial activity against *P. falciparum* [225]. Zamamidines A (**347**) and B (**348**) displayed cytotoxic activity against P388 cells (IC_50_ = 13.8 and 14.8 µg/mL, respectively) [217]. 

Finally, an interesting example of a 1,1′-hybrid manzamine dimer Kauluamine (**349**), isolated from the sponge *Prianos* sp. (Figure 70), revealed moderate immunosuppressive effect in a mixed lymphoma reaction [243].

##### General Syntheses of β-Carboline Alkaloids

Within the last decade, the synthesis of βCs has been quite extensively reviewed from diverse perspectives, focusing on the construction of the 9*H*-pyrido[3,4-*b*]indole [244]. Some of these authors distinguished between classical and current approaches, and a brief summary of each is provided below.

Classical routes, summarized in Figure 71 and Figure 72, are mostly dominated by the use of acid-/base-catalyzed or photochemical metal-free approaches. The most commonly exploited synthetic route for the formation of the βCs core, even nowadays, is the Pictet–Spengler reaction (Figure 71, method A) [245], starting from readily available tryptophan derivatives and carbonyl compounds. Another variation of this method includes the in situ reduction of nitriles (Figure 71, method B) [246]. A third variation of this methodology is the Bilschler–Napieralski reaction (Figure 71, method C) [247], in which amido-trypthophan derivatives are converted to electrophilic chlorimines using P_2_O_5_ or POCl_3_. All three routes yield tetrahydro-βC derivatives (THβCs), which require further oxidation steps to generate dihydro-βC (DHβC) or βCs. An important feature of the Pictet–Spengler approach for the synthesis of saturated carbolines is the possibility of inducing chirality by employing enantioselective aid catalysts [245].

Other early works reported the synthesis of βCs from 3-vinylindoles (Figure 72, method A) [248], Diels–Alder reactions (Figure 72, method B) [249], the Pd-catalyzed intramolecular arylation of anilinobromopyridines (Figure 72, method C) [250], Graebe–Ullmann reactions (Figure 72, method D) [251], the intramolecular nucleophilic substitutions of anilinofluoropyridines (Figure 72, method E) [252], and the photocyclization of anilinopyridines (Figure 72, method F) [253]. However, some of these procedures lacked functional group tolerance, forming only simple βC structures.

Over the past two decades, the number of chemical tools for organic synthesis has grown exponentially and, given the promising application of βCs as a drug, several new methodologies have been developed to build its azacarbazol skeleton. Mordi and Arshad performed an extensive review of these new methodologies [254], grouping them into the following categories: Larock heteroannulation (Figure 73A), C-H activation reactions (Figure 73B), Cycloaddition reactions (Figure 73C), 6π-Electrocyclizations (Figure 73D), Electrophilic cycloaromatization (not reported for βC so far), Cross-coupling reactions (Figure 73E), and Radical nucleophilic substitution (Figure 73F). Summarizing all of these processes is a difficult quest, given the wide range of chemical structures that could be potential starting materials and the transformations reported. Therefore, only one example of each is represented in Figure 73.

In this scenario, the elaboration of these scaffolds remains a hot area of research, although classical approaches are still preferred in most drug discovery programs. Notably, the development of valuable synthetic intermediates through these methodologies has allowed us to also explore a great number of further derivatization processes [255].

#### 2.3.4. Other Annelated Indole Alkaloids

In this section, some examples of annelated indole alkaloids (**350**–**354**), with varied structures, have been included due to their cytotoxic activity against several human cancer cell lines (Table 2).

## 3. Conclusions

Marine Indole Alkaloids comprise a wide variety of families of compounds. They originate from numerous marine organisms, such as fungi, sponges, corals, and mollusks, among others. As they are compounds released in order to survive against pathogens/predators in their own natural environment, they have important biological and pharmacological properties, such as antibacterial (potentially interesting to combat resistance from hospital bacteria) and anticancer (to avoid the resistance that some patients develop against certain therapies). Likewise, they have been shown to be potentially useful for treating certain eating disorders and diabetes. In this sense, MIAs can be considered as potential MDR modulators and/or sources of promising lead compounds, as demonstrated by the antibacterial and anticancer properties of some MIAs shown in this review. However, despite these promising applications, around 86% of MIAs’ potential remains largely underexplored, probably due to the absence of a systematic approach for exploring their pharmacological activity at clinically relevant concentrations for drug discovery. To harness the full therapeutic potential of MIAs, it is imperative to develop new bioassay techniques and synthetic protocols. These innovations would enable the precise interrogation of MIAs and facilitate their straightforward modification to enhance pharmacological efficacy. Although some MIAs may initially exhibit biological inactivity, strategic chemical modifications hold promise for optimizing their pharmacological properties. We believe that these approaches could represent a critical advancement in the quest for novel therapies to address current and emerging diseases, particularly in the face of challenges posed by antibiotic-resistant superbugs and therapy-resistant cancers.

## Figures and Tables

**Figure 1 marinedrugs-22-00126-f001:**
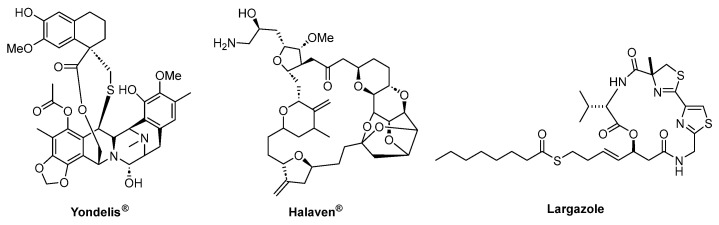
Example of marine drugs accepted by the FDA.

**Figure 2 marinedrugs-22-00126-f002:**
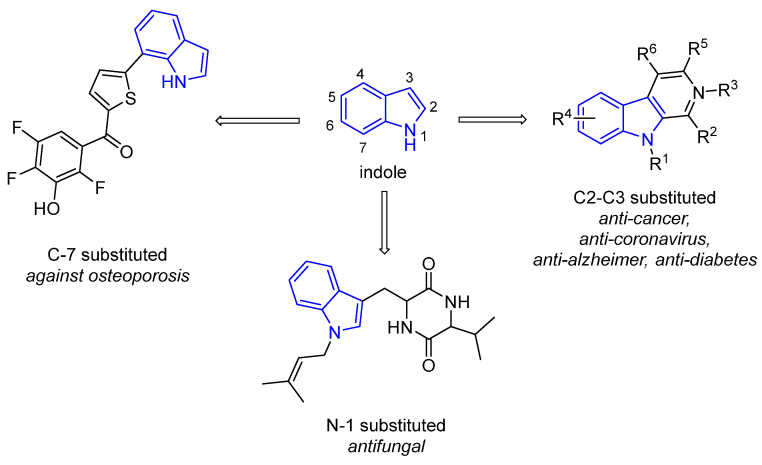
Structure/activity relationships of indole derivatives.

**Figure 3 marinedrugs-22-00126-f003:**
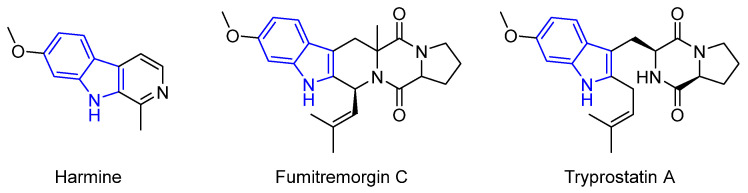
Structure of some marine indole alkaloids with anticancer activity.

**Figure 4 marinedrugs-22-00126-f004:**
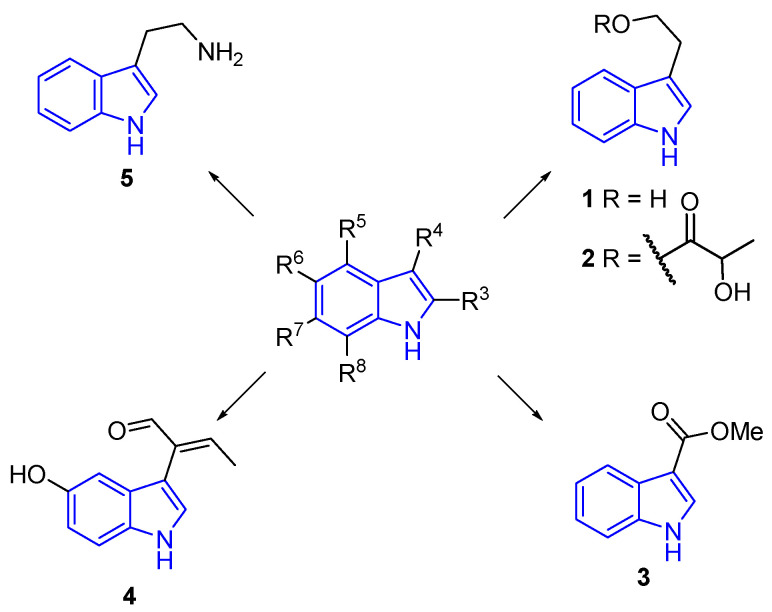
Structures of C3-acyclic substituted SIAs **1**–**5**.

**Figure 5 marinedrugs-22-00126-f005:**
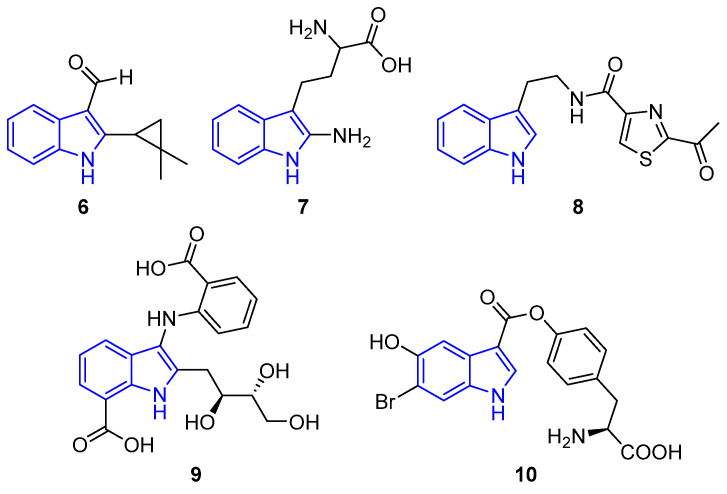
Structures of C3-carbaldehyde/carboxy-substituted SIAs **6**–**10**.

**Figure 6 marinedrugs-22-00126-f006:**
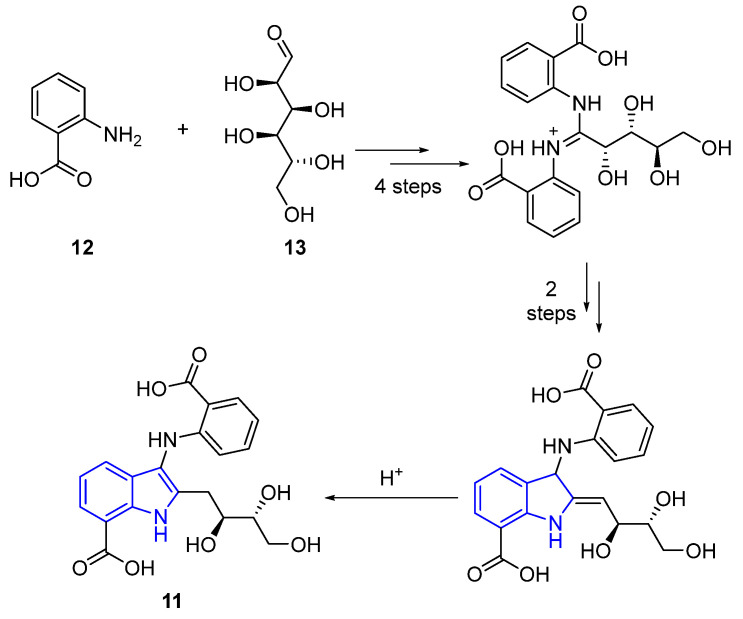
Biogenetic route to obtain Anthranoside C (**11**).

**Figure 7 marinedrugs-22-00126-f007:**
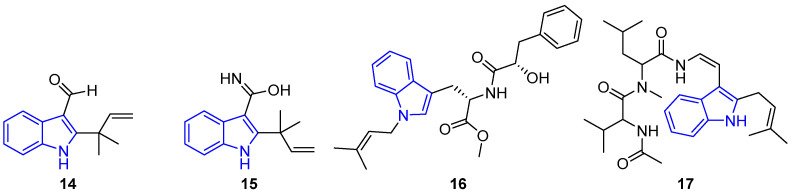
Structures of prenylated SIAs **14**–**17**.

**Figure 8 marinedrugs-22-00126-f008:**
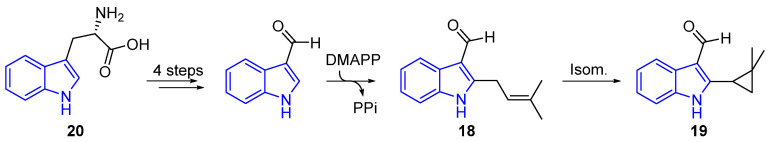
Biogenesis of isomers **18** and **19**.

**Figure 9 marinedrugs-22-00126-f009:**
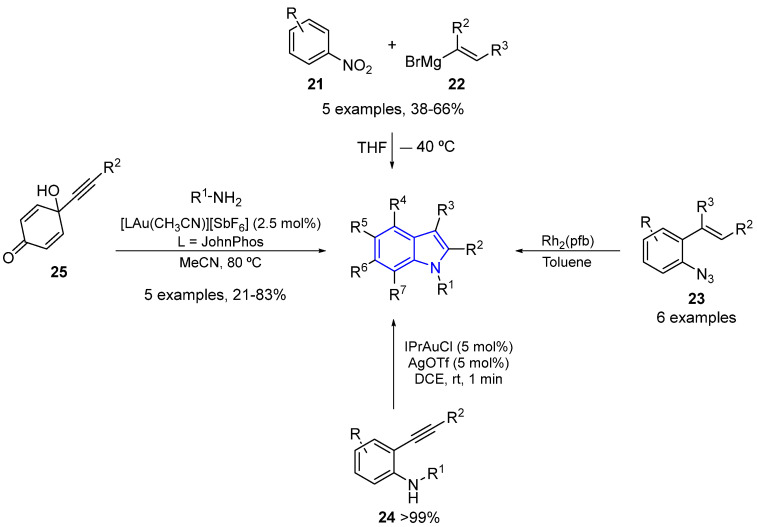
Synthetic approaches in the preparation of functionalized SIAs.

**Figure 10 marinedrugs-22-00126-f010:**
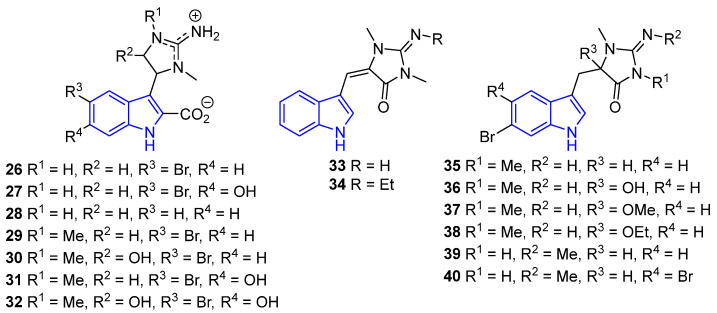
Trachycladindole A–G (**26**–**32**), aplysinopsins (**33**), and their derivatives (**34**–**40**).

**Figure 11 marinedrugs-22-00126-f011:**
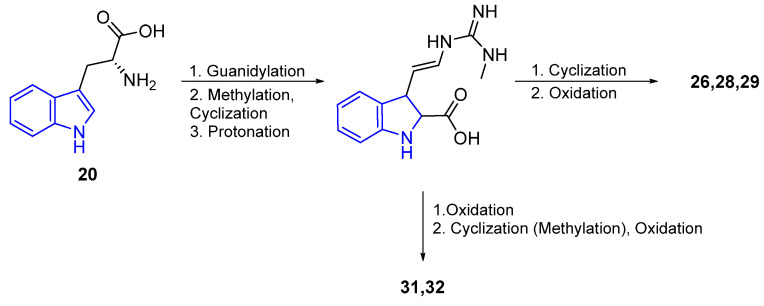
Trachycladindole hypothetical biosynthesis by A. Hentz.

**Figure 12 marinedrugs-22-00126-f012:**
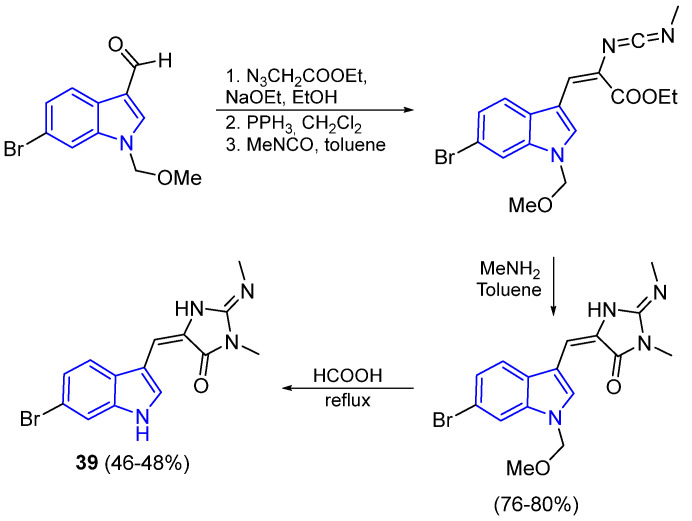
Stanovnik and Svete’s synthesis of Aplysinopsin derivate **39**.

**Figure 13 marinedrugs-22-00126-f013:**
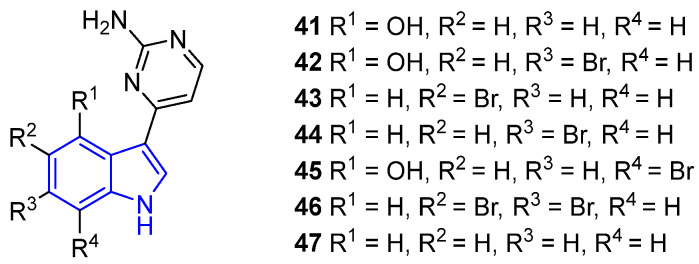
Structure of Meridianins A–G (**41**–**47**).

**Figure 14 marinedrugs-22-00126-f014:**
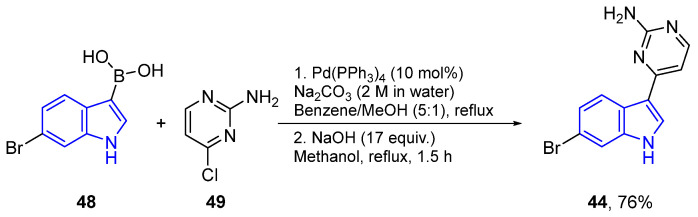
Jing and Yang synthesis of Meridianin D (**44**).

**Figure 15 marinedrugs-22-00126-f015:**
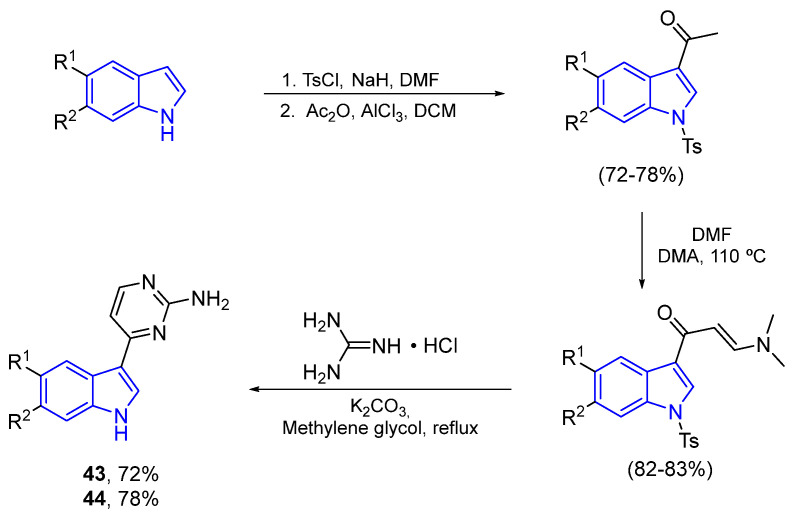
Fresneda and Molina’s synthesis of Meridianins C (**43**) and D (**44**).

**Figure 16 marinedrugs-22-00126-f016:**
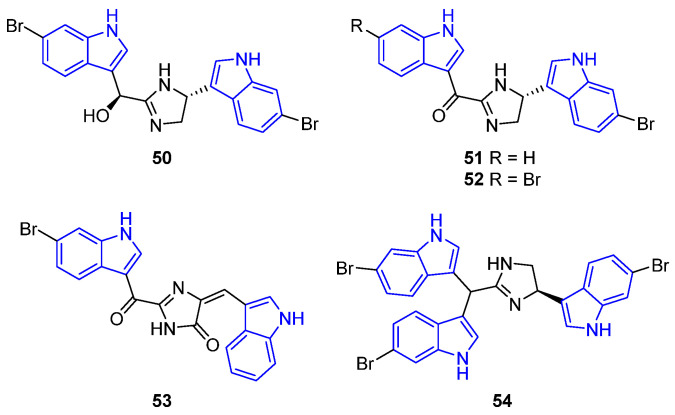
Structure of bis-indoles **50**–**53** and tris-indole **54**.

**Figure 17 marinedrugs-22-00126-f017:**
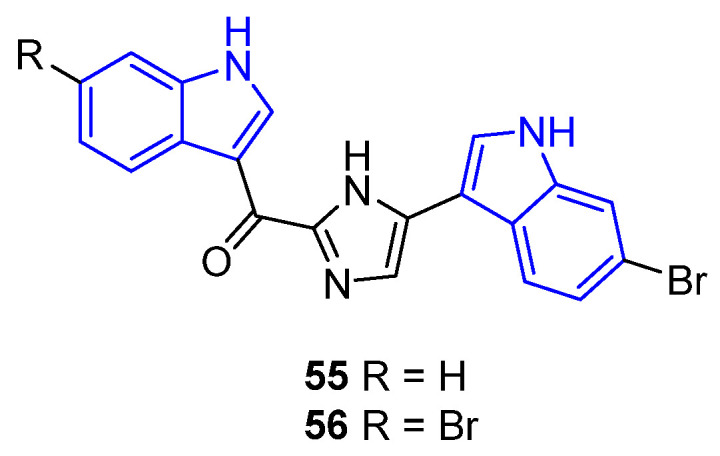
Structure of bromodeoxytopsentin (**55**) and dibromodeoxytopsentin (**56**).

**Figure 18 marinedrugs-22-00126-f018:**
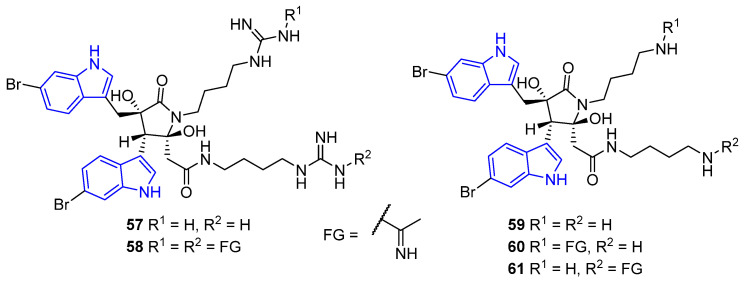
Structure of eusynstelamides A–B and D–F (**57**–**61**).

**Figure 19 marinedrugs-22-00126-f019:**
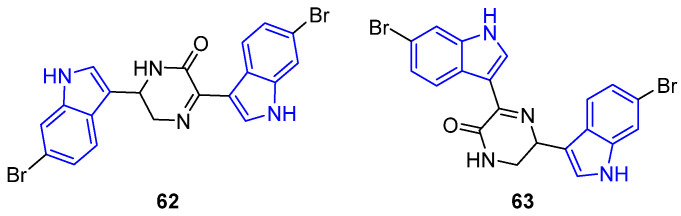
Structure of Hamacanthins A–B (**62**–**63**).

**Figure 20 marinedrugs-22-00126-f020:**
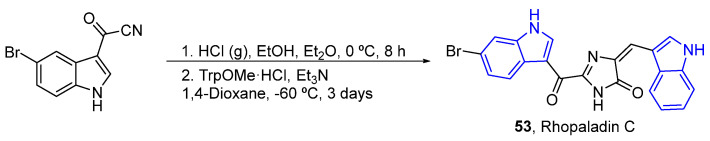
Janosik et al. [64] synthesis of Rhopaladin C (**53**).

**Figure 21 marinedrugs-22-00126-f021:**
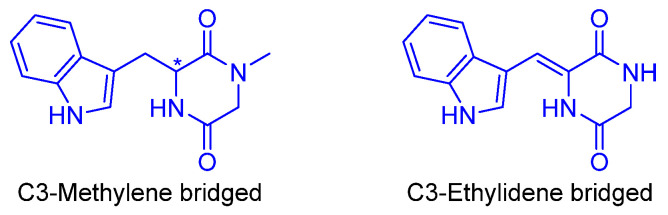
Basic structures of Simple Diketopiperazines.

**Figure 22 marinedrugs-22-00126-f022:**
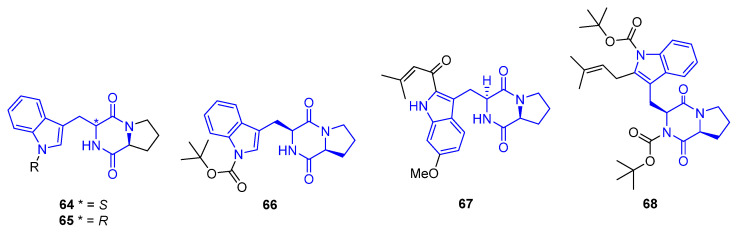
Chemical structures of simple DKPs **64**–**68** with C3-methylene bridge.

**Figure 23 marinedrugs-22-00126-f023:**
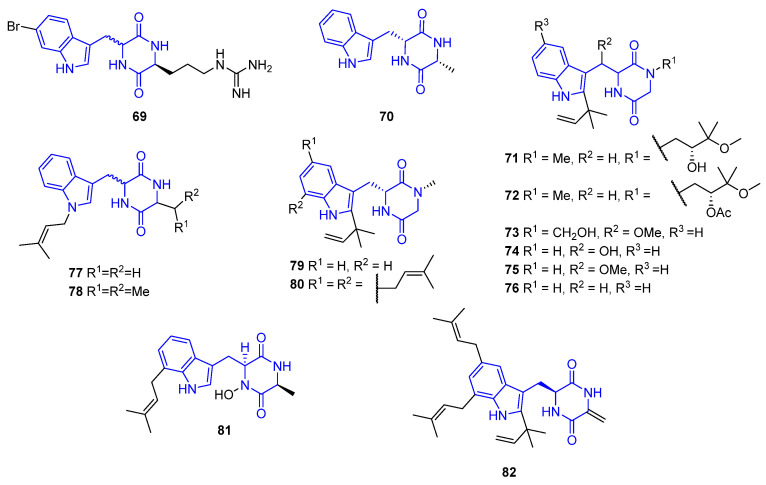
Chemical structures of simple DKPs **69**–**82** with C3-methylene bridge.

**Figure 24 marinedrugs-22-00126-f024:**
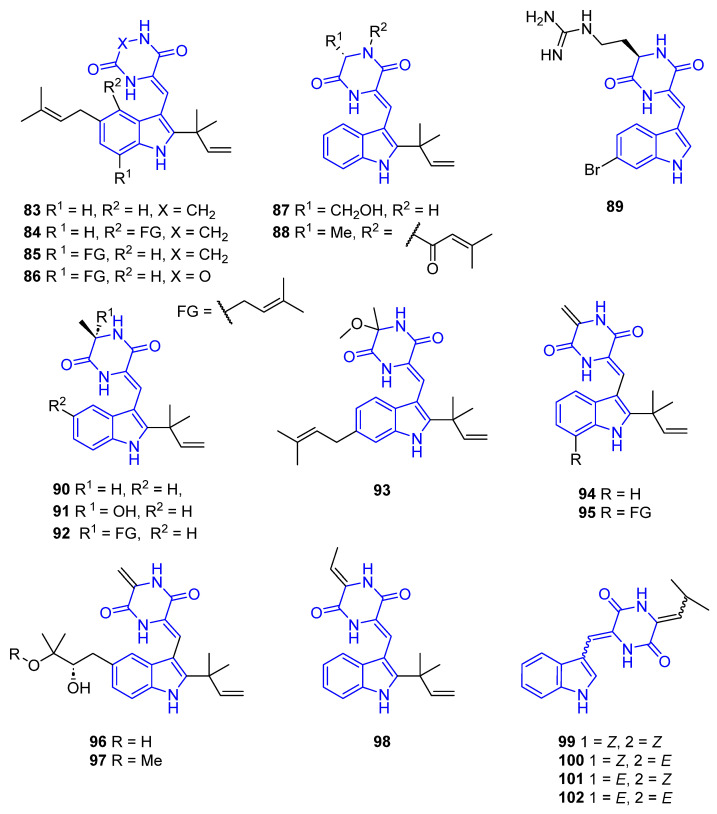
Chemical structures of simple DKPs **83**–**102** with C3-ethylidene bridge.

**Figure 25 marinedrugs-22-00126-f025:**
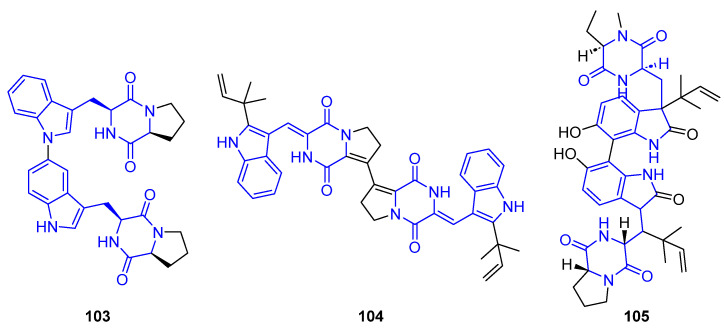
Chemical structures of bis-indol DKPs **103**–**105**.

**Figure 26 marinedrugs-22-00126-f026:**
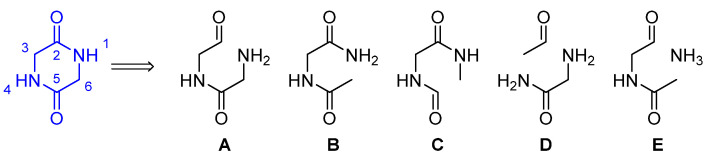
Possible disconnections of a 2,5-diketopiperazine ring.

**Figure 27 marinedrugs-22-00126-f027:**
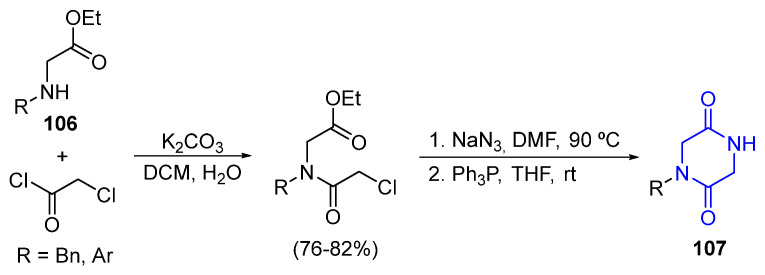
Aza-Wittig cyclization to synthesize DKPs **107**.

**Figure 28 marinedrugs-22-00126-f028:**
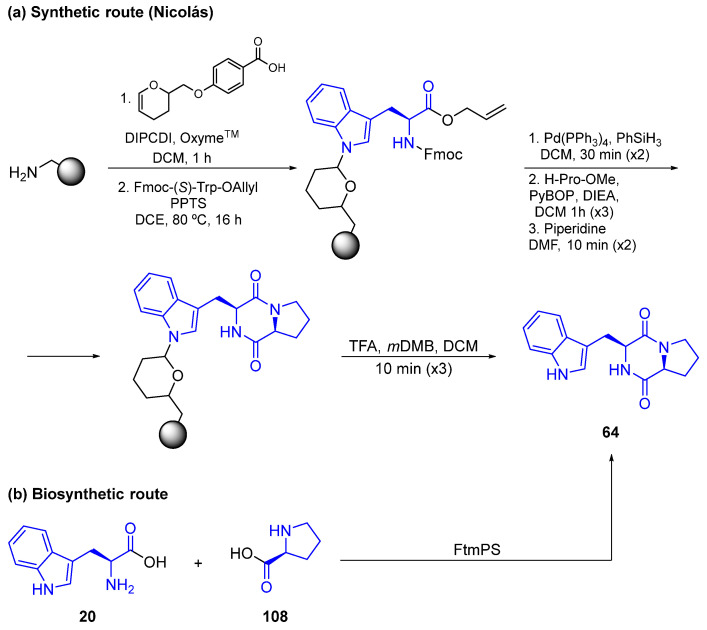
Synthesis and biosynthesis of Brevianamide F (**64**).

**Figure 29 marinedrugs-22-00126-f029:**
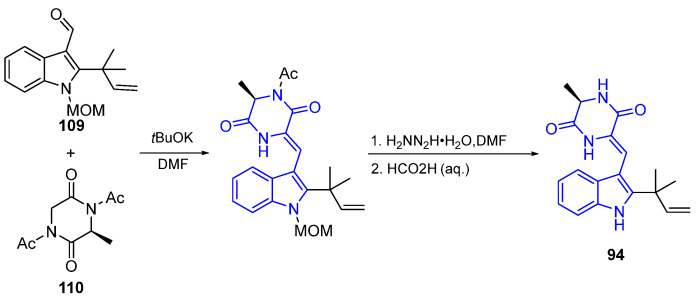
Synthesis of Neochenulin A (**94**).

**Figure 30 marinedrugs-22-00126-f030:**
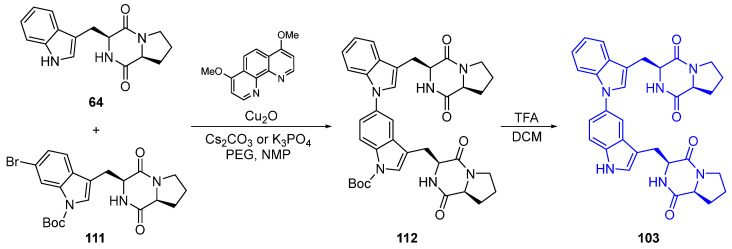
Synthesis of Aspergilazine A (**103**).

**Figure 31 marinedrugs-22-00126-f031:**
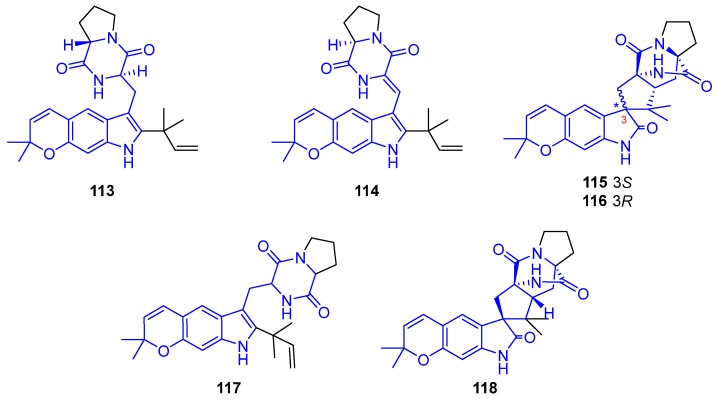
Structures of compounds **113**–**118**.

**Figure 32 marinedrugs-22-00126-f032:**
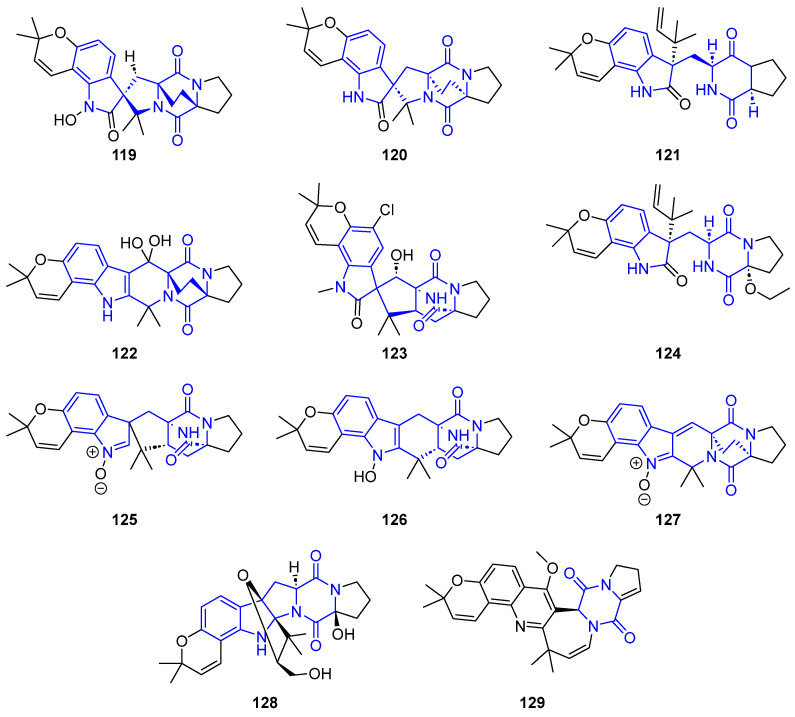
Structures of compounds **119**–**129**.

**Figure 33 marinedrugs-22-00126-f033:**
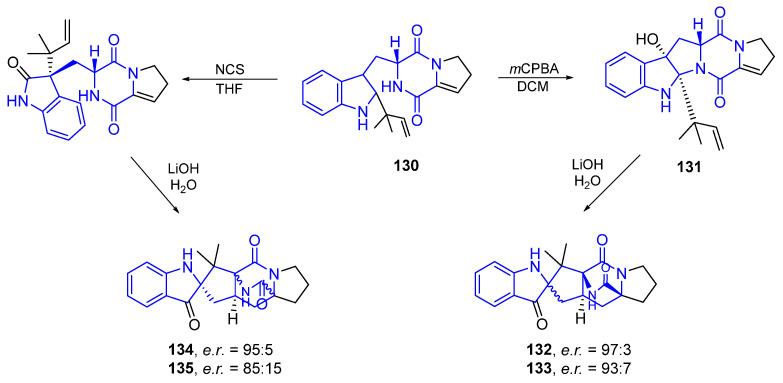
Example of synthesis of breviamides **132**–**135**.

**Figure 34 marinedrugs-22-00126-f034:**
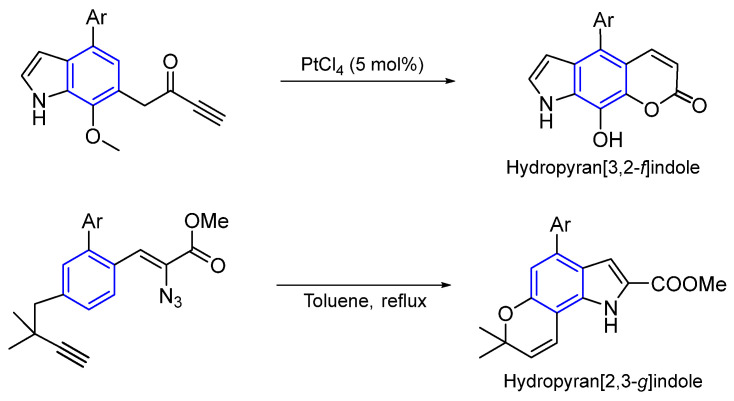
Synthesis of dimethyhydropyranoindole nucleus.

**Figure 35 marinedrugs-22-00126-f035:**
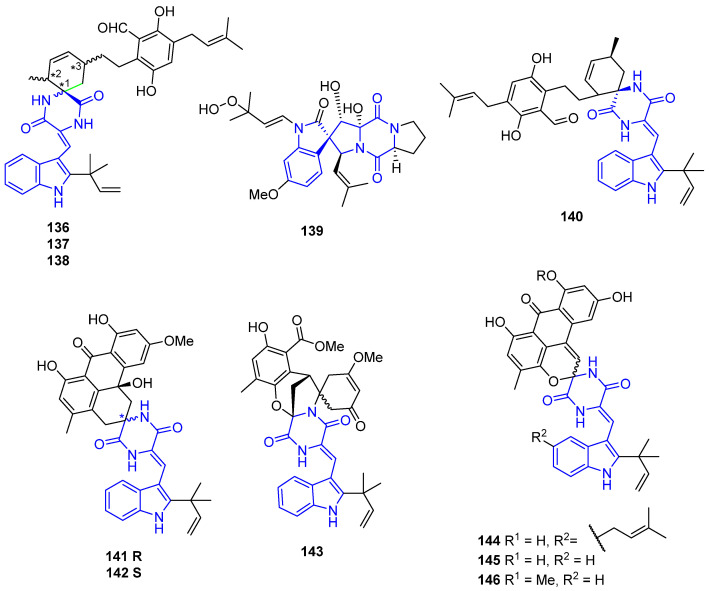
Structures of compounds **136**–**146**.

**Figure 36 marinedrugs-22-00126-f036:**
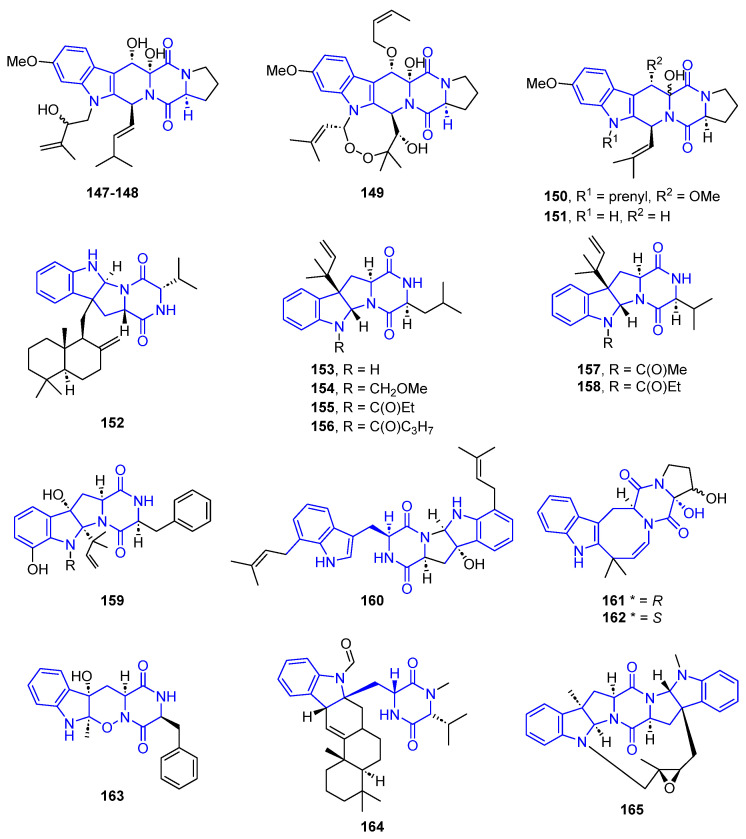
Structures of compounds **147**–**165**.

**Figure 37 marinedrugs-22-00126-f037:**
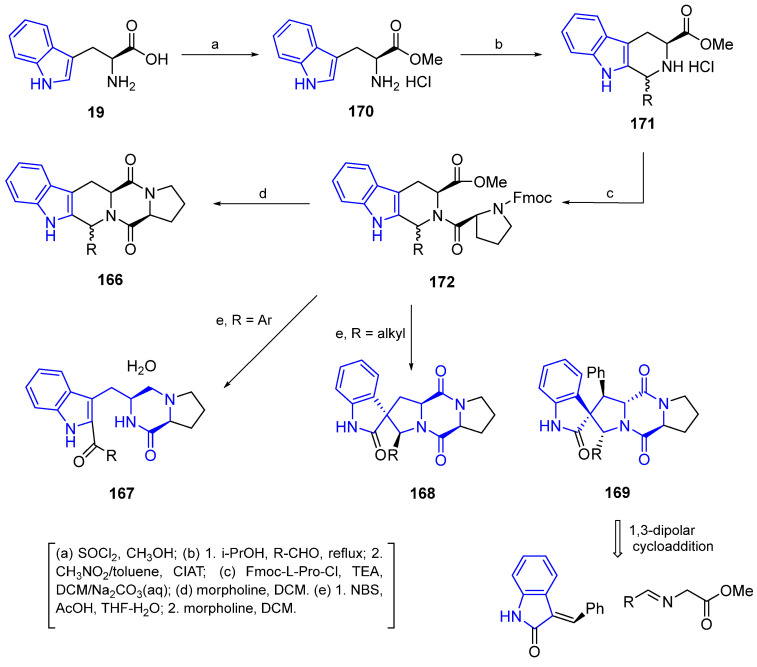
Synthesis of Indole diketopiperazine alkaloids.

**Figure 38 marinedrugs-22-00126-f038:**
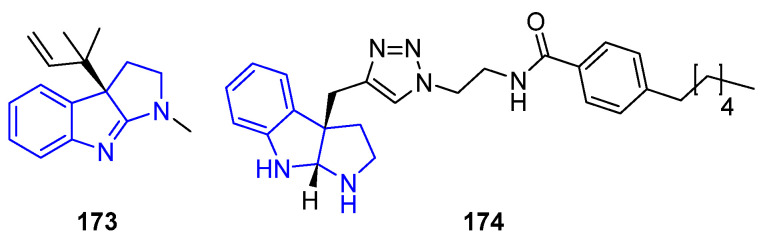
Structures of compounds **173** and **174**.

**Figure 39 marinedrugs-22-00126-f039:**
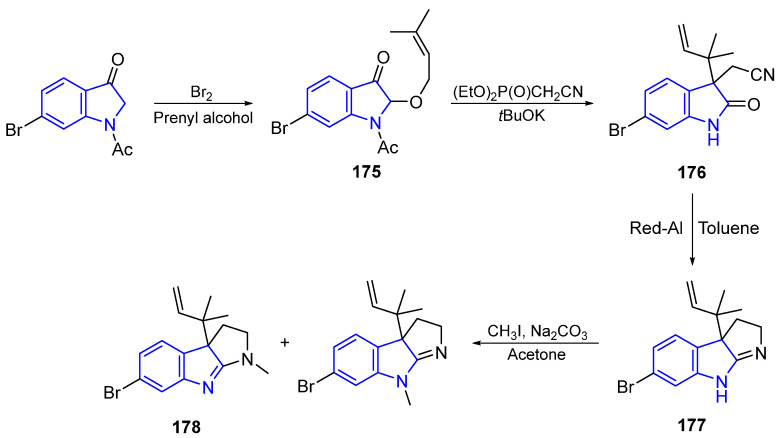
Classical synthesis of Flustramine C (**178**).

**Figure 40 marinedrugs-22-00126-f040:**
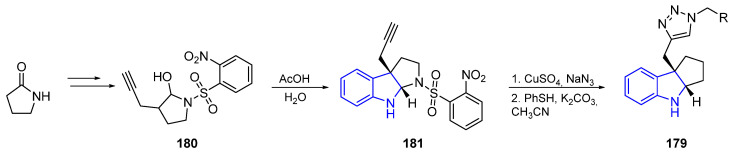
Synthesis of the flustramines analogs **179**.

**Figure 41 marinedrugs-22-00126-f041:**
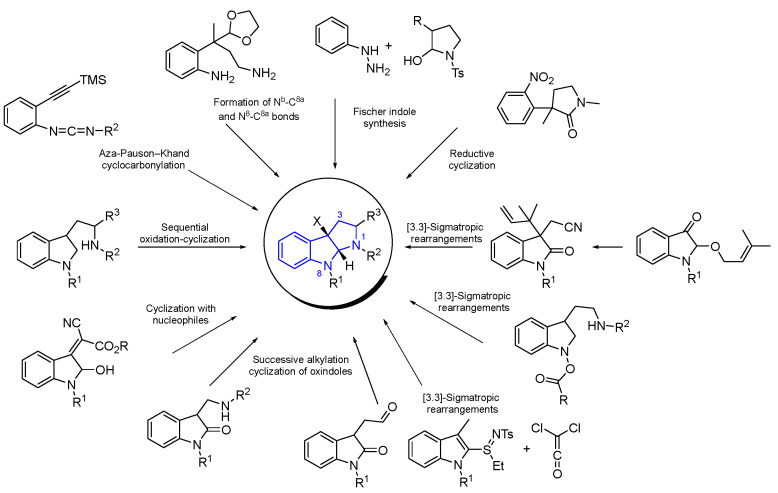
Synthetic routes of tricyclic HPI.

**Figure 42 marinedrugs-22-00126-f042:**
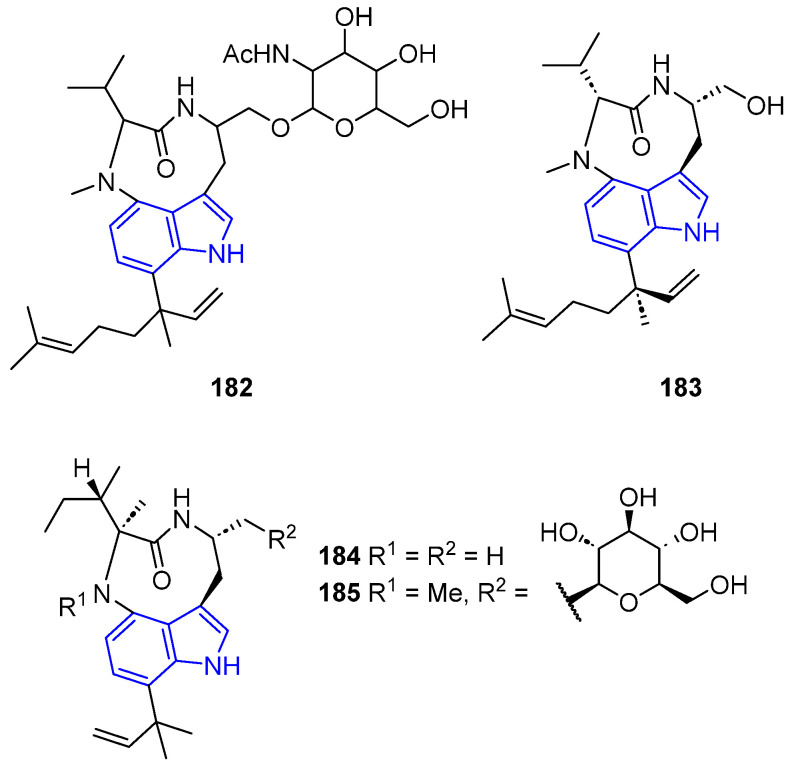
Structures of compounds **182**–**185**.

**Figure 43 marinedrugs-22-00126-f043:**
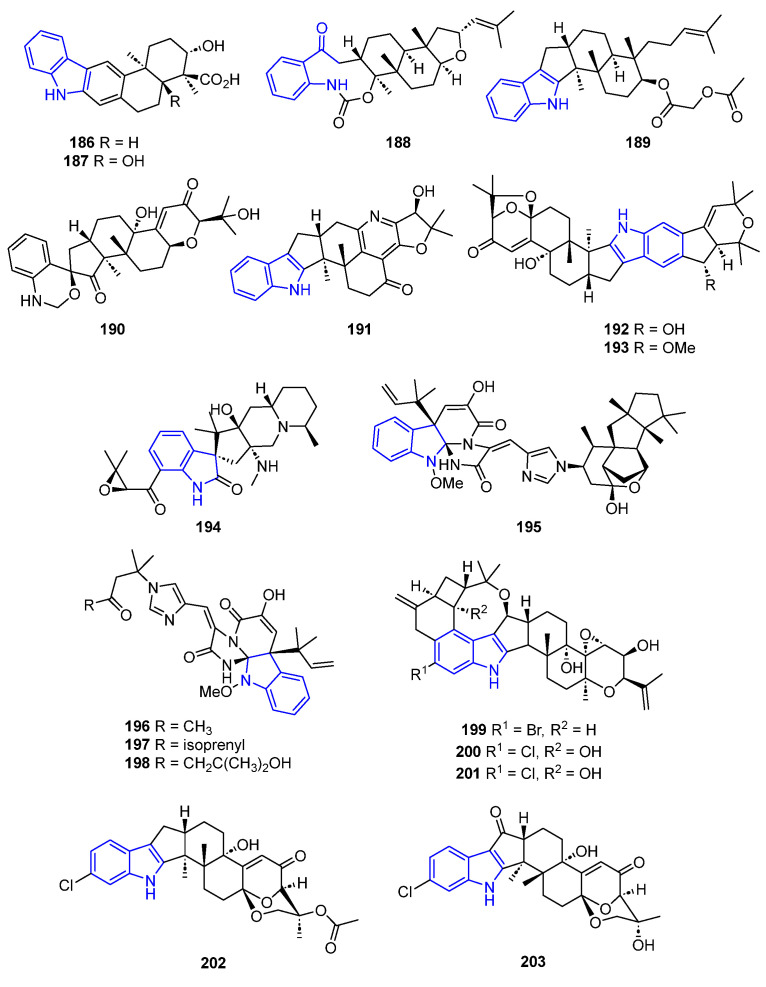
Structures of compounds **186**–**203**.

**Figure 44 marinedrugs-22-00126-f044:**
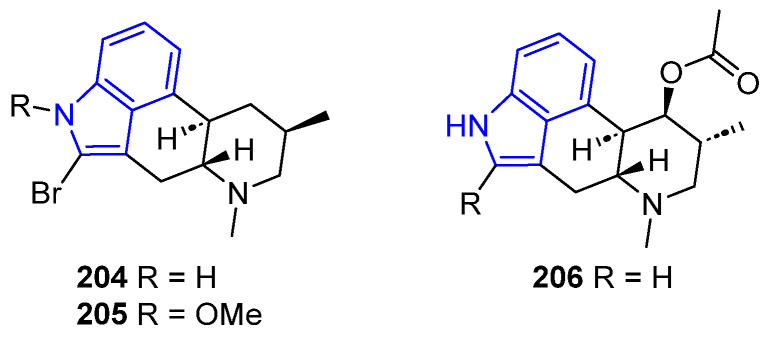
Structures of compounds **204**–**206**.

**Figure 45 marinedrugs-22-00126-f045:**
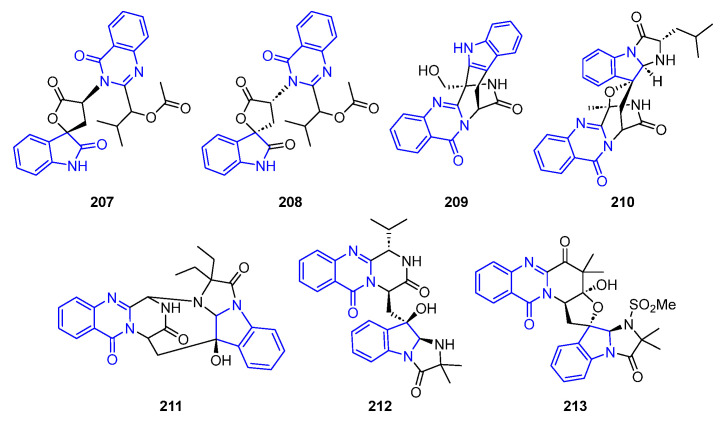
Structures of compounds **207**–**213**.

**Figure 46 marinedrugs-22-00126-f046:**
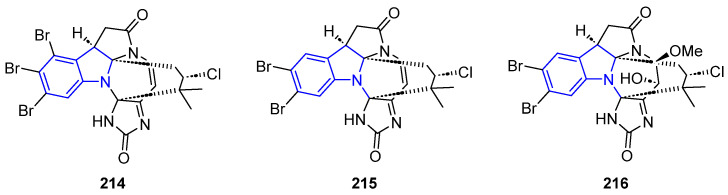
Structures of compounds **214**–**216**.

**Figure 47 marinedrugs-22-00126-f047:**
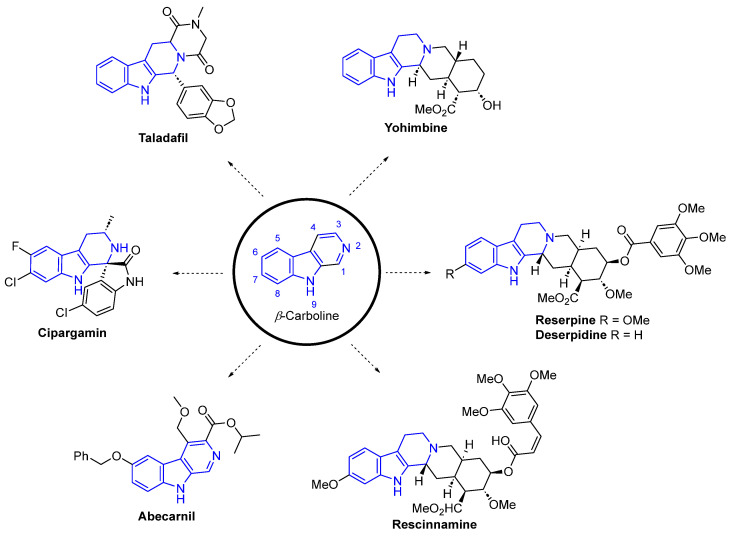
Representative commercialized β-carboline drugs.

**Figure 48 marinedrugs-22-00126-f048:**
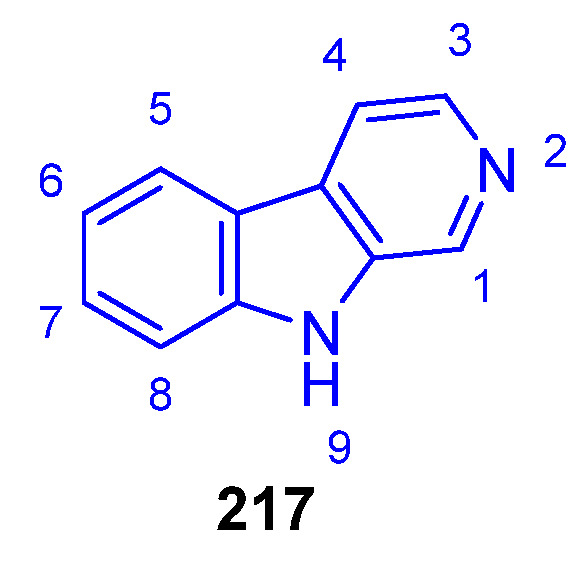
Structure of Norharmane **217**.

**Figure 49 marinedrugs-22-00126-f049:**
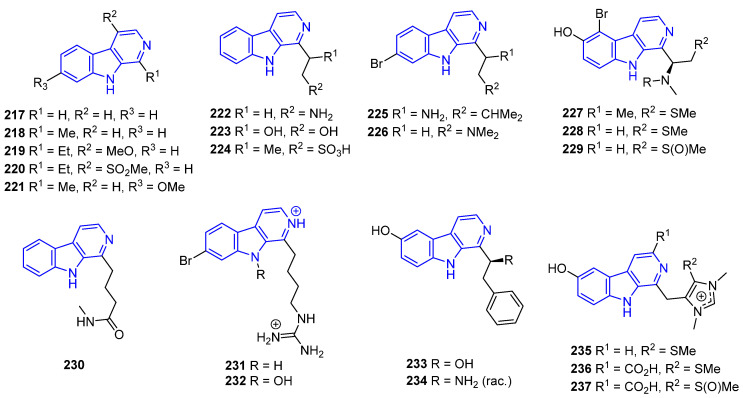
C1-substituted βC compounds **217**–**237**.

**Figure 50 marinedrugs-22-00126-f050:**
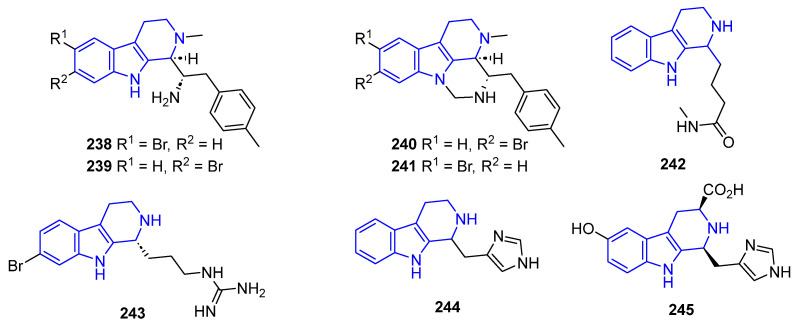
C1-substituted βC compounds **238**–**245**.

**Figure 51 marinedrugs-22-00126-f051:**
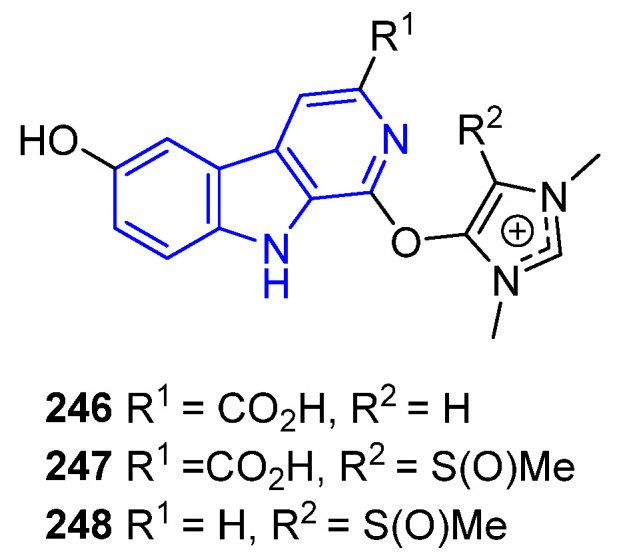
C1-substituted βC compounds **246**–**248**.

**Figure 52 marinedrugs-22-00126-f052:**
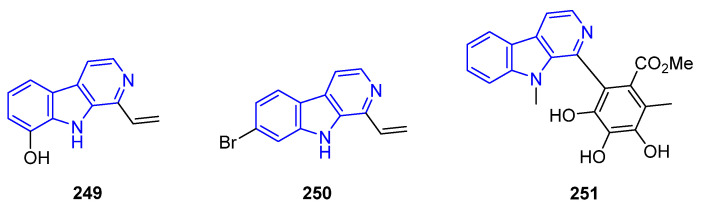
C1-substituted βC compounds **249**–**251**.

**Figure 53 marinedrugs-22-00126-f053:**
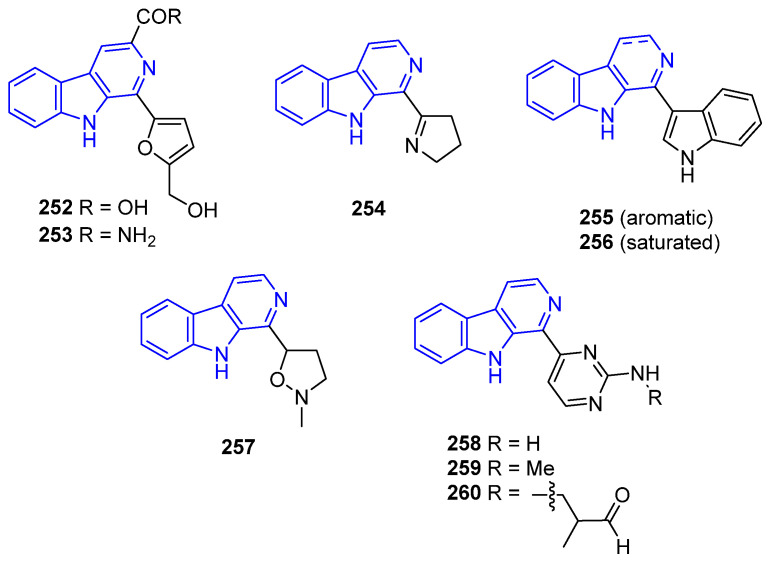
C1-substituted βC compounds **252**–**260**.

**Figure 54 marinedrugs-22-00126-f054:**
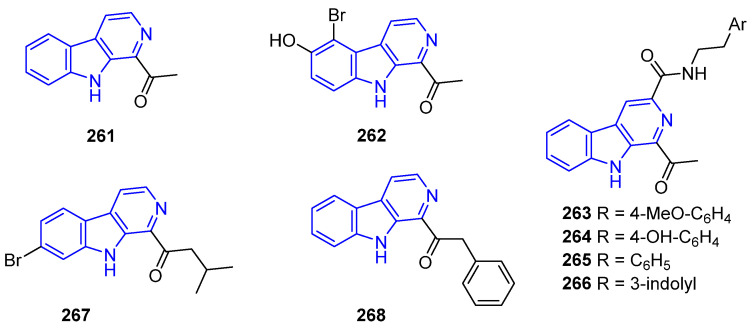
C1-substituted βC compounds **261**–**268**.

**Figure 55 marinedrugs-22-00126-f055:**
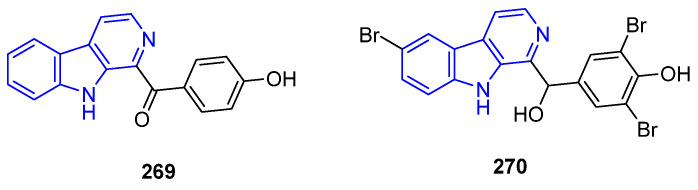
C1-substituted βC compounds **269**–**270**.

**Figure 56 marinedrugs-22-00126-f056:**
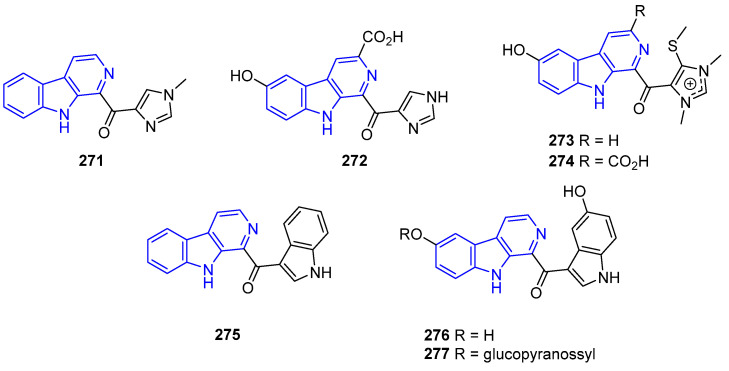
C1-substituted βC compounds **271**–**277**.

**Figure 57 marinedrugs-22-00126-f057:**
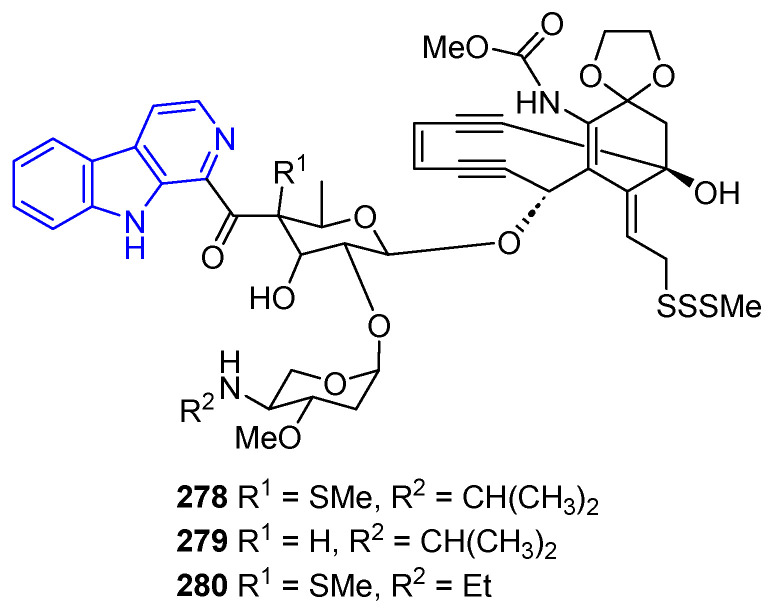
Chemical structure of Shishijimicin A–C (**278**–**280**).

**Figure 58 marinedrugs-22-00126-f058:**
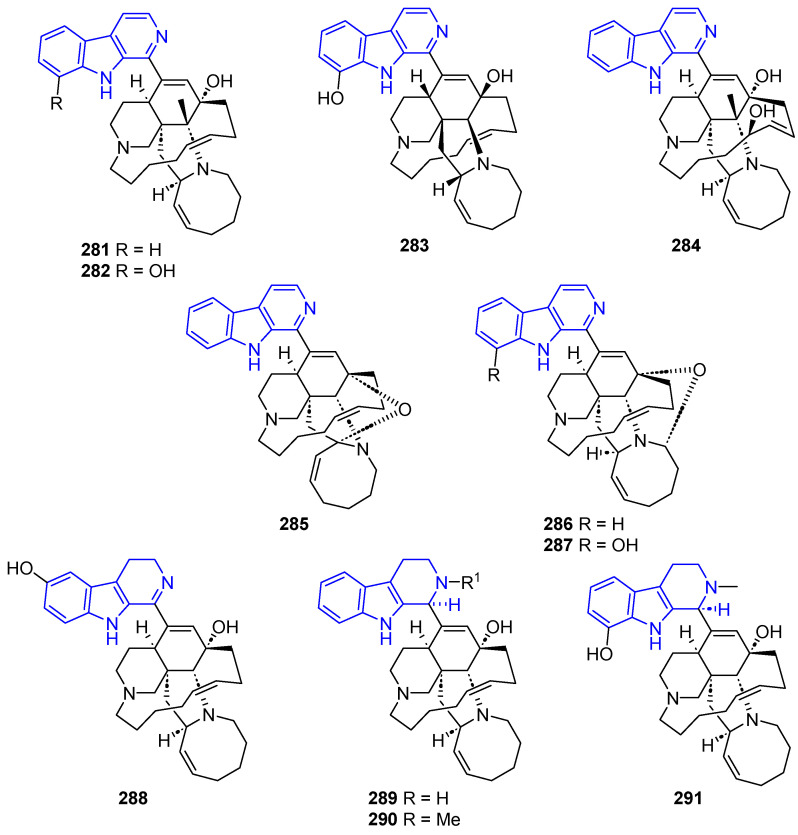
Chemical structures of Manzamines **281**–**291**.

**Figure 59 marinedrugs-22-00126-f059:**
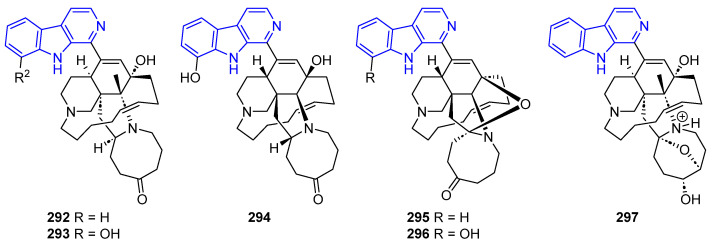
Chemical structures of Manzamines **292**–**297**.

**Figure 60 marinedrugs-22-00126-f060:**
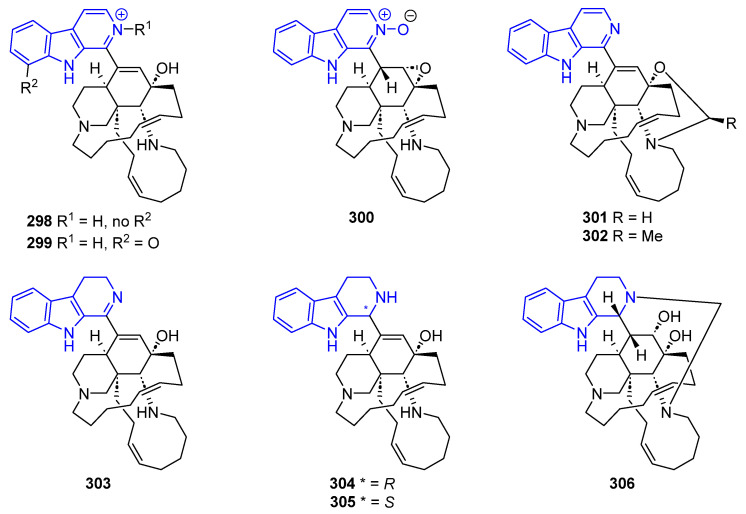
Chemical structures of Manzamines **298**–**306**.

**Figure 61 marinedrugs-22-00126-f061:**
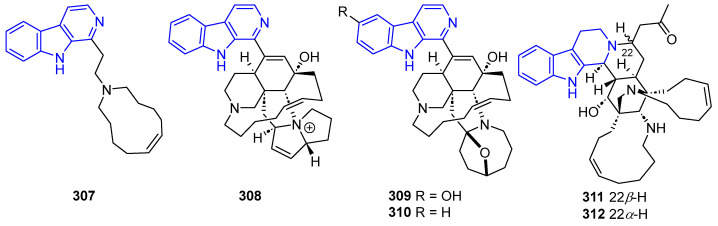
Chemical structures of Manzamines **307**–**312**.

**Figure 62 marinedrugs-22-00126-f062:**
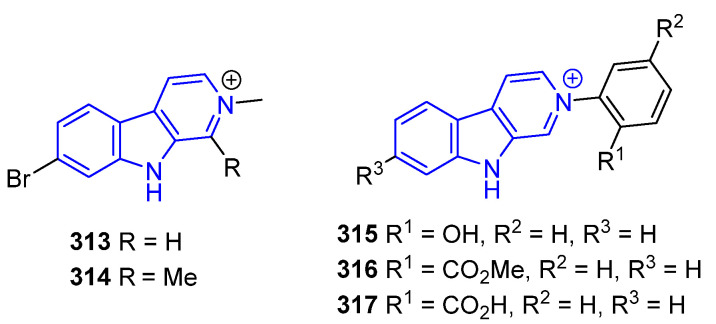
N2-substituted βC compounds **313**–**317**.

**Figure 63 marinedrugs-22-00126-f063:**
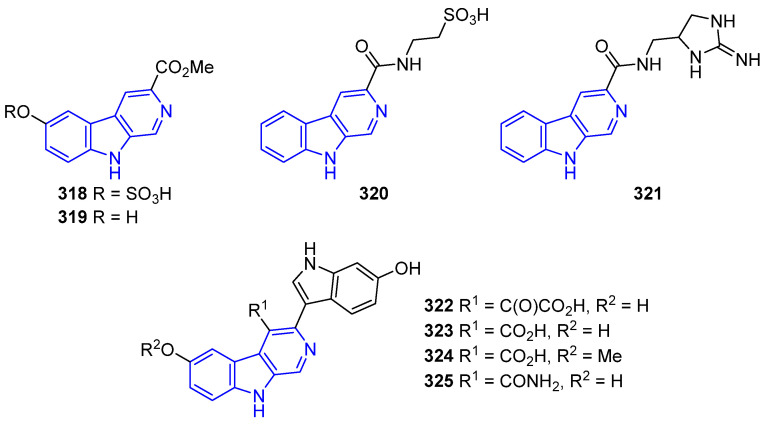
C3-substituted βC compounds **318**–**325**.

**Figure 64 marinedrugs-22-00126-f064:**
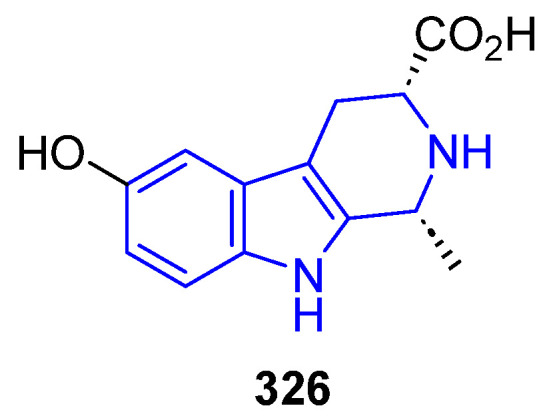
Chemical structures of Hyrtioerectine B (**326**).

**Figure 65 marinedrugs-22-00126-f065:**
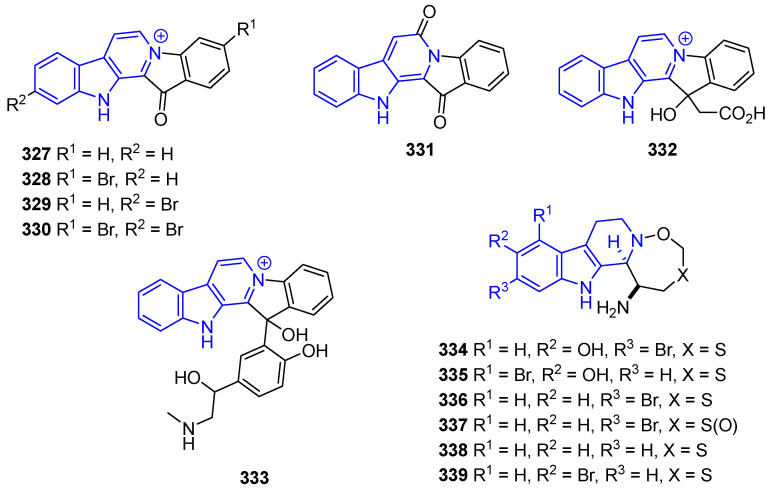
Annelated C1-N2 βC compounds **327**–**339**.

**Figure 66 marinedrugs-22-00126-f066:**
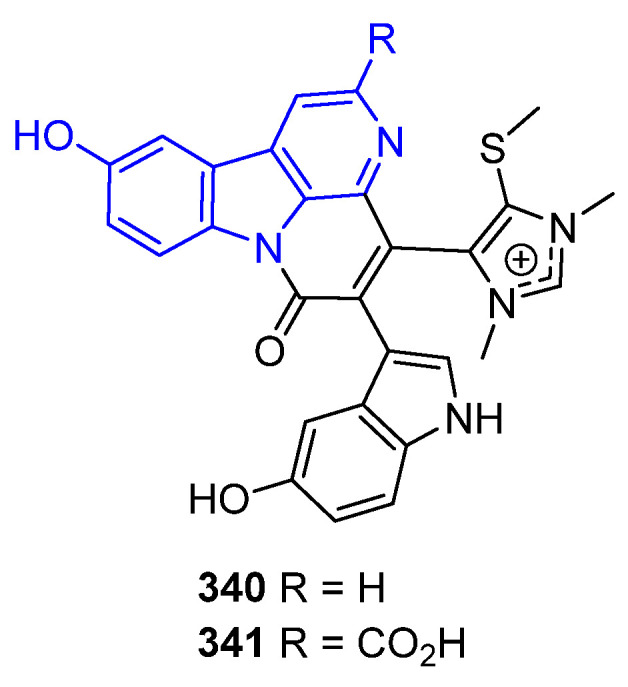
Annelated βC compounds **340**–**341**.

**Figure 67 marinedrugs-22-00126-f067:**
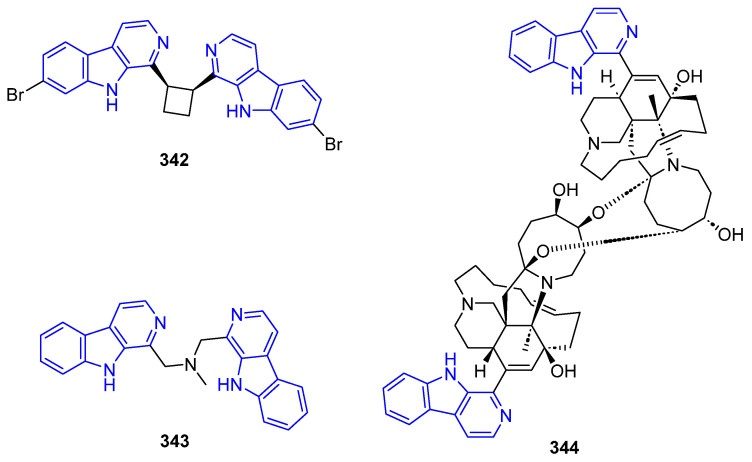
Naturally occurring marine βC 1,1-dimers **342**–**344**.

**Figure 68 marinedrugs-22-00126-f068:**
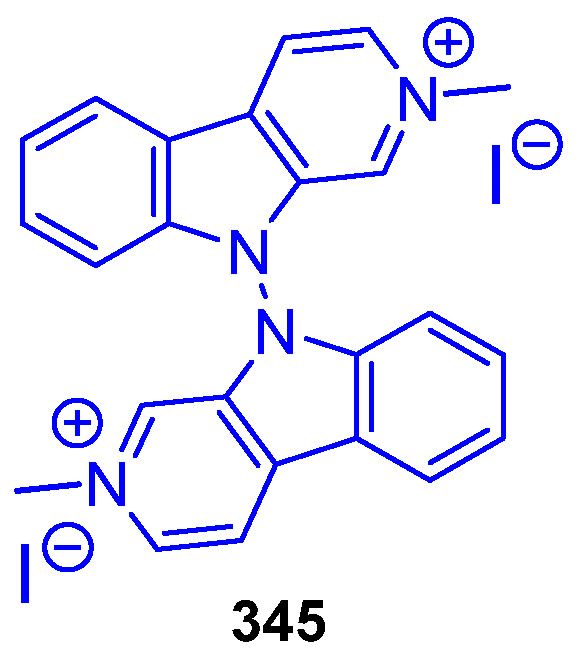
Structure of marine βC 9,9-dimer **345**.

**Figure 69 marinedrugs-22-00126-f069:**
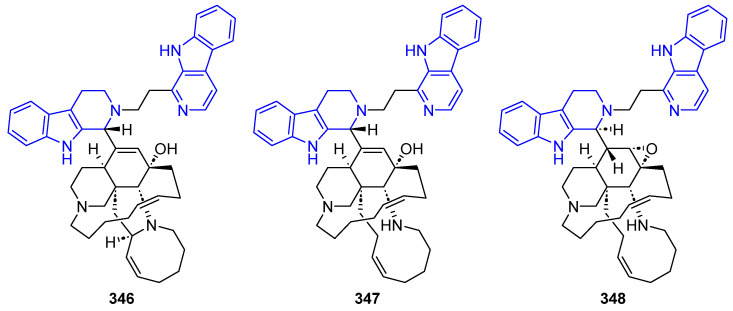
Structure of manzamine hybrid dimers **346**–**348**.

**Figure 70 marinedrugs-22-00126-f070:**
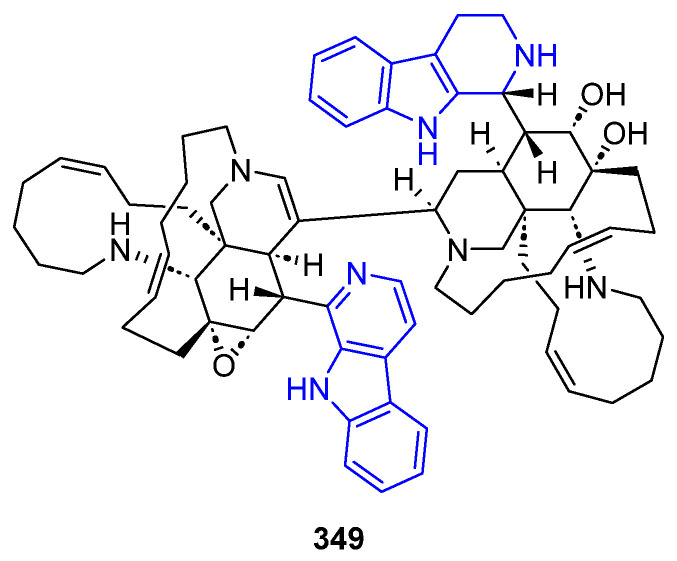
Structure of Kauluamine (**349**).

**Figure 71 marinedrugs-22-00126-f071:**
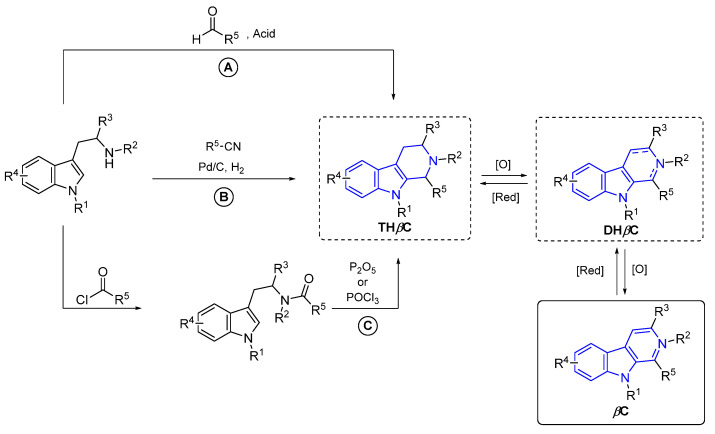
Most employed synthetic routes for synthesizing βCs.

**Figure 72 marinedrugs-22-00126-f072:**
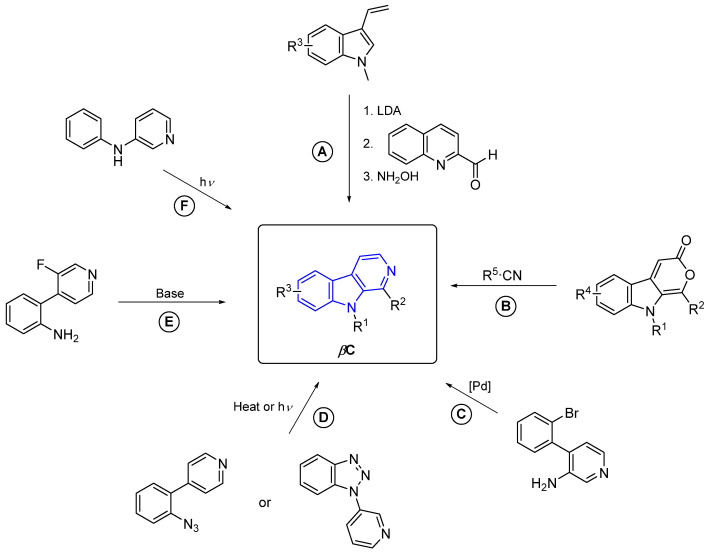
Other classical general synthetic routes towards the synthesis of βCs.

**Figure 73 marinedrugs-22-00126-f073:**
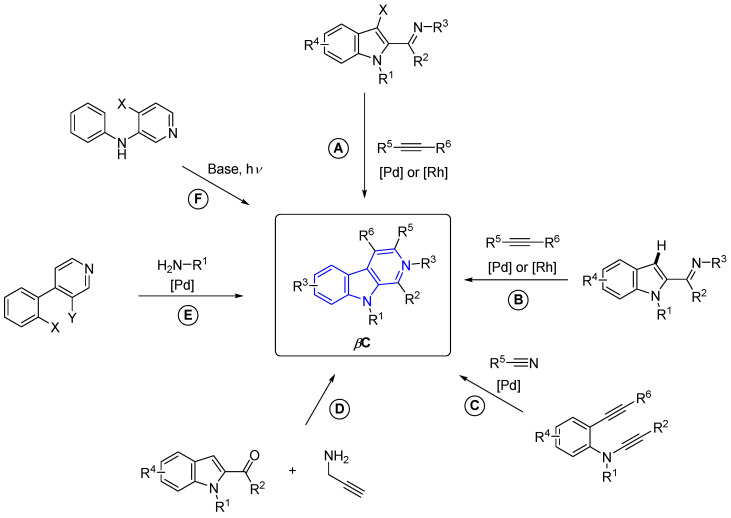
Representative modern approaches towards the synthesis of βCs.

**Table 1 marinedrugs-22-00126-t001:** Bioactivity of Meridianins A–G (**41**–**47**).

Meridianin	Anticancer Effects	Prevention of Alzheimer’s Disease	Antimalarial Effects	Antitubercular Effects
A (**41**)	Hela	GSK-3β, CK1δ, Dyrk1A and CLK1 ^2^	*P. falciparum*	nd ^1^
B (**42**)	PTP, Hep2, U937, LMM3		nd	nd
C (**43**)	PTP, Hep2, HT29, RD, U937, LMM3, Hela, MDA-MB-231, A549		*P. falciparum*	*M. tuberculosis*
D (**44**)	PTP, Hep2, HT29, RD, U937, LMM3, Hela, A549		nd	*M. smegatis * ^3^
E (**45**)	PTP, Hep2, U937, LMM3	nd	nd	nd
F (**46**)	Hep2, U937, LMM3	nd	nd	nd
G (**47**)	Hela	Dyrk1A	*P. falciparum*	*M. tuberculosis*

^1^ nd: not determined, ^2^ Inhived kinases, ^3^ Antibiofilm activity.

**Table 2 marinedrugs-22-00126-t002:** Annelated indole alkaloids (**350**–**354**) and their cytotoxic activity.

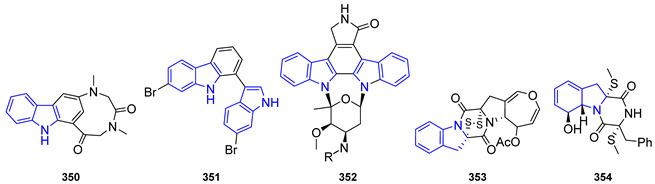			
Name Annelated Indol	Structure	Cytotoxicity [Reference]	[Reference]
Antipathine A	**350**	SGC-791, Hep-G2	[256]
Bromine Indolyl-carbazoles	**351**	HL-60, HeLa	[257]
Staurosporines	**352**	MV4-11	[258]
Deoxyapoaranotin	**353**	HCT-116 ^1^	[259]
Phomazine B	**354**	HL-60, HCT-116, K562, MGC-803, A549 ^1^	[260]

^1^ Via apoptosis-inducing effects.

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
