# Peer review of "Current Status of Indole-Derived Marine Natural Products: Synthetic Approaches and Therapeutic Applications"

_marinedrugs, 2024, doi:10.3390/md22030126_

Round 1

Reviewer 1 Report

Comments and Suggestions for Authors

The review article by Arroyo et al covers primarily the synthesis and biological activities of indole-derived marine natural products. The biosynthesis of some of these molecules is also presented and discussed. Given the unusual structural features and therapeutic potential of several such compounds, the present review is likely to capture the interests of a number of synthetic and medicinal chemists.

In general, literature coverage is thorough, and diagrams are of high quality. However, the English needs to be improved in some cases and corrected throughout the manuscript. Some aspects, both grammatical and statement of fact are either poorly presented or just not easy to understand. The list of comments and corrections provided below is by no means exhaustive. Otherwise, this is a good review worthy of publication after the appropriate changes/corrections have been made to the manuscript.

- Title: Change “ Current Scenario of Indole Marine-derivatives: ” to  “ Indole-derived Marine Natural Products: ” or “ Current Status of Indole-derived Marine Natural Products: ”

- Page 1, line 29: “better tolerance in general by patients” - I wonder if this is true. Are there any credible references? 

- Page 2, line 42:  Change “being indole derivatives one of the most promising” to “with indole derivatives being one of the most promising”

- Page 5, line 146:  Change “May Zin et al. developed a biogenesis of isomers compounds 17 and 18,” to “May Zin et al. proposed a biogenesis of isomeric compounds 17 and 18,”

- Page 7, line 187:  Change “Aplysinopsins derivate” to “Aplysinopsin derivate”

- Page 11, line 286: …. in the design of novel…

- Page 12, line 311: Change “Other two examples whose estructure derived from the hexahydropyrrolopyrazine 311 are…” to  “Another two examples whose structures are derived from hexahydropyrrolopyrazine 311 are…”

- Page 12, lines 315-316: “, being this activity strongly enhanced due to the C2-isoprene and…” to “ . This activity was strongly enhanced due to the C2-isoprene and…”

- Page 12, line 330:  ….obtained from the M-3 strain…

- Page 13, line 353:  ….a brominated cyclodipeptide isolated from…

- Page 16, line 413:  Figure 28. Synthesis and biosynthesis of Brevianamide F (63). 

- Page 16, line 426: Figure 28. Synthesis and biosynthesis of Brevianamide F (63). Do not use the word “Scheme” in other Figure captions everywhere.

- Page 18, line 456:  Notoamides are a large family containing…

- Page 19, lines 485-486: The synthesis starts with the preparation of (+)-Dehydro-deoxybreviamide E….in a five-step gram-scale procedure.

- Page 19, lines 486-487: …followed by exposure of…

- Page 19, line 489:  Treatment of 130 with NCS and then LiOH provided Brevianamide…

- Page 20, line 495:  As seen, some marine indole alkaloids have a hydropyrano ring …

- Page 22, line 546:  …containing an eight-membered…

- Page 22, line 551:  …containing an oxindole ring…

- Page 22, line 558:  …isolated from a marine…

- Page 22, line 566: A general strategy…

- Page 22, line 567:  Three types of analogues…

- Page 29, lines 718-719:   Either remove or correct this part of the sentence: “being the applications of the dimers closely related to their parent monomers”

- Page 34, line 877-878:   …displays a great variety of activities…

- Page 40, lines 1022-1023:  Some recent research has shown a potential trend in which the….dimers tend to be more active than the corresponding monomers [158].

- Page 41, line 1046:  It is worth mentioning that…

- Page 42, line 1082:  … use of…

- Page 42, line 1085:  Other variations of this method include….

- Page 43, line 1100:  ...lacked functional group tolerance…

Comments on the Quality of English Language

See comments in the previous section.

Author Response

In first place, I sincerely appreciate the valuable feedback provided to improve the quality of the paper. The reviewer insights are greatly valued, and I am committed to incorporating them effectively into the manuscript. In this sense, all the spelling mistakes and tense inconsistencies have been diligently corrected. As for the remaining comments and suggestions, I will address each point individually in separate responses.

- Title: Change “Current Scenario of Indole Marine-derivatives: ” to  “ Indole-derived Marine Natural Products: ” or “ Current Status of Indole-derived Marine Natural Products: ”

               Response: In response to your suggestion, we have updated the title to better reflect the content of the paper. It now reads: "Current Status of Indole-derived Marine Natural Products: Synthetic Approaches and Therapeutic Applications."

- Page 1, line 29: “better tolerance in general by patients” - I wonder if this is true. Are there any credible references? 

               Response: Thank you for bringing up this concern. Upon reviewing the statement regarding "better tolerance in general by patients," we acknowledge the importance of substantiating such claims with credible references. During the comprehensive research for this work, we came across multiple examples for which the cytotoxic activity was exclusively exerted towards cancer human cells, but not against non-malignant ones; or in which less side effects were observed. However, we recognise that we may have been overstated the generality of this trend. Thus, we have decided to remove the sentence from the manuscript.

- Page 2, line 42:  Change “being indole derivatives one of the most promising” to “with indole derivatives being one of the most promising”

               Response: The sentence has been changed in the manuscript.

- Page 5, line 146:  Change “May Zin et al. developed a biogenesis of isomers compounds 17 and 18,” to “May Zin et al. proposed a biogenesis of isomeric compounds 17 and 18,”

               Response: The sentence has been changed in the manuscript.

- Page 7, line 187:  Change “Aplysinopsins derivate” to “Aplysinopsin derivate”

               Response: The sentence has been changed in the manuscript.

- Page 11, line 286: …. in the design of novel…

Response: “design” has been corrected in the manuscript.

- Page 12, line 311: Change “Other two examples whose estructure derived from the hexahydropyrrolopyrazine 311 are…” to  “Another two examples whose structures are derived from hexahydropyrrolopyrazine 311 are…”

Response: The sentence has been changed in the manuscript.

- Page 12, lines 315-316: “, being this activity strongly enhanced due to the C2-isoprene and…” to “ . This activity was strongly enhanced due to the C2-isoprene and…”

               Response: The sentence has been changed in the manuscript.

- Page 12, line 330:  ….obtained from the M-3 strain…

Response: Past tense has been corrected.

 - Page 13, line 353:  ….a brominated cyclodipeptide isolated from…

               Response: “wich” has been removed.

  - Page 16, line 413:  Figure 28. Synthesis and biosynthesis of Brevianamide F (63).  - Page 16, line 426: Figure 28. Synthesis and biosynthesis of Brevianamide F (63). Do not use the word “Scheme” in other Figure captions everywhere.

               Response: Figure captions have been corrected: Figures: 27, 28, 29 and 30.

- Page 18, line 456:  Notoamides are a large family containing…

 Response: The sentence has been changed in the manuscript.

- Page 19, lines 485-486: The synthesis starts with the preparation of (+)-Dehydro-deoxybreviamide E….in a five-step gram-scale procedure.

 Response: The sentence has been changed in the manuscript.

- Page 19, lines 486-487: …followed by exposure of…

 Response: The sentence has been changed in the manuscript.

- Page 19, line 489:  Treatment of 130 with NCS and then LiOH provided Brevianamide…

 Response: The sentence has been changed in the manuscript.

- Page 20, line 495:  As seen, some marine indole alkaloids have a hydropyrano ring …

 Response: The sentence has been changed in the manuscript.

- Page 22, line 546:  …containing an eight-membered…

 Response: The sentence has been corrected in the manuscript.

- Page 22, line 551:  …containing an oxindole ring…

  Response: The sentence has been corrected in the manuscript.

- Page 22, line 558:  …isolated from a marine…

 Response: Past tense has been corrected.

- Page 22, line 566: A general strategy…

  Response: The sentence has been corrected in the manuscript.

 - Page 22, line 567:  Three types of analogues…

   Response: The sentence has been corrected in the manuscript.

- Page 29, lines 718-719:   Either remove or correct this part of the sentence: “being the applications of the dimers closely related to their parent monomers”

   Response: The sentence has been removed.

- Page 34, line 877-878:   …displays a great variety of activities…

   Response: The sentence has been changed in the manuscript

- Page 40, lines 1022-1023:  Some recent research has shown a potential trend in which the….dimers tend to be more active than the corresponding monomers [158].

Response: The sentence has been changed in the manuscript.

- Page 41, line 1046:  It is worth mentioning that…

               Response: The sentence has been changed in the manuscript.

- Page 42, line 1082:  … use of…

               Response: The sentence has been corrected in the manuscript.

- Page 42, line 1085:  Other variations of this method include….

               Response: The sentence has been corrected in the manuscript.

- Page 43, line 1100:  ...lacked functional group tolerance…

Response: The sentence has been corrected in the manuscript.

Reviewer 2 Report

Comments and Suggestions for Authors

This review article summarizes a recent topic of indole alkaloids isolated from marine organisms.  A huge number of indole alkaloids are listed and categorized based on their structural features.  Moreover, some synthetic schemes are described for selected important compounds.  

The well-organized information described in this review would be of interest for a number of researchers in natural product, medicinal, and synthetic chemistry.

I recommend this manuscript for publication in Marine Drugs.

 A few minor comments:

Page 1, line 32: The structure of Ziconotide is not given.  However, this compound is a natural polypeptide having a huge molecular structure.  Because of the less importance to mention about this compound, which is not an indole derivative, it can be deleted.

 Page 1, line 33: Eribulin is not a natural product but a synthetic derivative of Halicondrin B, an anticancer natural product.  But if the authors use the word “marine drugs” for both of natural product and artificial synthetic derivative, it is OK.

Author Response

In first place, I sincerely appreciate the valuable feedback provided to improve the quality of the paper. The reviewer insights are greatly valued, and I am committed to incorporating them effectively into the manuscript. In this sense, all the spelling mistakes and tense inconsistencies have been diligently corrected. As for the remaining comments and suggestions, I will address each point individually in separate responses.

-“Page 1, line 32: The structure of Ziconotide is not given.  However, this compound is a natural polypeptide having a huge molecular structure.  Because of the less importance to mention about this compound, which is not an indole derivative, it can be deleted”.

Response: We agree with your observation and have opted to remove the mention of Ziconotide from the text. As it is not an indole derivative and considering its substantial molecular structure, we agree that its inclusion is of lesser relevance to the focus of our discussion

- Page 1, line 33: Eribulin is not a natural product but a synthetic derivative of Halicondrin B, an anticancer natural product.  But if the authors use the word “marine drugs” for both of natural product and artificial synthetic derivative, it is OK

Response: In response to the query about Eribulin mesylate, we have chosen to retain its structure within the text referred as “Marine drug”. Our decision is based on the understanding that this term can encompass both natural products and synthetic derivatives in this context.

Reviewer 3 Report

Comments and Suggestions for Authors

The authors made huge efforts to summarize the indole containing natural products with marine origin. The review shows the chemical structures, the most interesting biological activities and the main synthetic routes leading to the most important scaffolds.

The manuscript is suitable for publiction after a minor revision.

Line 54. anti-Alzheimer’s disease

Line 58: C7 position

Line 66: harmine

Line 139 „in” is unnecessary

Line 163: 2.1.2. C3-(Iminoimidazolidin- and pyrazin-)-subtituted simple indole alkaloids

Line 209: pyrimidinyl

Figure 21. Basic structures of simple diketopiperizines

Line 330: The diketopiperazine 78, obtained

Line 409 and Figure 27: Aza-Wittig reaction

Figure 30: Compounds 111 and 112 are missing

Line 443: hydropyran[3,2-f]indole

Line 446: dimethylpyranoindole

Line456: hydropyran[3,2-e]indole

Figure 34: one oxygen is missing.

Line 558: Nocardioazine A (165) isolated

Line 559: displayed

Line 570: methyl L-tryptophan hydrochloride (capital letter is not necessary)

Line 572: provided

Line 582: Hexahydropyrrolo[2,3-b]indol (HPI) derivatives

Line 592: hexahydropyrrolo[2,3-b]indol

Line 701: 2.3.3. β-Carbolines

Line 759: 1-ethyl-4-methylsulfone-β-carboline (capital letter is not necessary)

Line 877: presents

Line 880: 6-O-(β-glucopyranosyl)hyrtiosulawesine

Line 881: 277…. has antimalarial activity

Line 1055: N-methylated

Line 1088: All these three routes

Comments on the Quality of English Language

Line 877: Hyrtiosulawesine (276), found in the Indonesian sponge Hyrtios erectus, presents 

Line 1088: All these three routes (instead of All this three...)

Author Response

In first place, I sincerely appreciate the valuable feedback provided to improve the quality of the paper. The reviewer insights are greatly valued, and I am committed to incorporating them effectively into the manuscript. In this sense, all the spelling mistakes and tense inconsistencies have been diligently corrected. As for the remaining comments and suggestions, I will address each point individually in separate responses.

- Line 54: “anti-Alzheimer’s disease” has been corrected.

- Line 58: “C7 position” has been corrected.

- Line 66: “harmine” has been corrected.

- Line 139: “in” has been removed.

- Line 163: 2.1.2. “C3-(Iminoimidazolidin- and pyrazin-)-subtituted simple indole alkaloids” has been corrected.

- Line 209: “pyrimidinyl” has been corrected.

- Figure 21. “Basic structures of simple diketopiperizines” has been corrected.

- Line 330: “The diketopiperazine 78, obtained” has been corrected.

- Line 409 and Figure 27: “Aza-Wittig reaction” have been corrected

- Figure 30: “Compounds 111 and 112 are missing”. Response: mistake with the numbers has been corrected in the figure 30.

- Line 443: “hydropyran[3,2-f]índole” has been corrected.

- Line 446: “dimethylpyranoindole” has been corrected.

- Line456: “hydropyran[3,2-e]índole” has been corrected.

- Figure 34: “one oxygen is missing”. Response: Double bond C=O has been added.

- Line 558: “Nocardioazine A (165) isolated” has been corrected.

- Line 559: “displayed” has been corrected.

- Line 570: “methyl L-tryptophan hydrochloride” (Methyl capital letter has been removed)

- Line 572: “provided” has been corrected.

- Line 582: “Hexahydropyrrolo[2,3-b]indol (HPI) derivatives.” (pyrrolo capital letter has been corrected).

- Line 592: “hexahydropyrrolo[2,3-b]indol” (pyrrolo capital letter has been corrected).

- Line 701: “2.3.3. β-Carbolines” (β-Carbolines capital letter has been corrected).

- Line 759: “1-ethyl-4-methylsulfone-β-carboline” (ethyl capital letter has been corrected)

- Line 877: “displays” has been added.

- Line 877: “Hyrtiosulawesine (276), found in the Indonesian sponge Hyrtios erectus, presents” has been corrected. 

- Line 880: 6-O-(β-glucopyranosyl)hyrtiosulawesine

- Line 881: “277…. has antimalarial activity” has been corrected

- Line 1055: “N-methylated” Response: methyl groups have been included in the figure 68.

- Line 1088: “All these three routes” has been corrected

Reviewer 4 Report

Comments and Suggestions for Authors

Marine-originating indole alkaloids have various applications, primarily in pharmaceutical and biomedical fields due to their diverse pharmacological properties. Some of these applications include: drug discovery, anti-microbial, anti-cancer, anti-inflammatory, anti-malarial and anti-viral agents, neuroprotective agents, and biomedical research and development/drug candidates.

This review summarizes the structural diversity, synthetic (or semi-synthetic) strategies for obtaining such chemical structures, and some of their biomedical applications. I believe it is a good, comprehensive overview of the state-of-the-art, but it could be better structured. Right now, it consists of three very disproportionate chapters. A reasonably-sized introduction, a short conclusion and a very large chapter 2, which is the bulk of the review. I would have preferred to see a more balanced distribution of material across different chapters, as announced in the Abstract: first an overview of structural classes, then synthetic methodologies, and finally therapeutic applications. This way the review will be easily perused by both organic chemists interested likely more in the former and biologists/biochemists/pharmacologists interested rather more in the latter aspects.

Also, as a comprehensive review, I was surprised to find that staurosporine, a promising anticancer compound from this class, was not mentioned. See, for example: https://www.frontiersin.org/journals/microbiology/articles/10.3389/fmicb.2022.957473/full , or other similar literature sources.

Also, tryptophol, a sleep-inducing compound isolated from marine sponge Ircinia spiculosa could have been mentioned as well.

Also, although I am not sure they have been discovered in marine species to date, I wonder if any secodine alkaloids could be included also as part of the structural diversity outlined in the review. Consider adding these as well if any recent literature on marine sources exists.

Minor issues:

Title: no need for hyphenation; "Marine Derivatives" would be just fine

Line 5: London (-on missing)

Line 4: comma needed before "Daniel"

Line 7: Science (misspelled)

Line 26: were shown (not "have shown")

Line 42: word order incorrect; rephrase to "one of the most promising being indole derivatives"

Line 66: indole (spelling issue)

Line 94: consist (no "-ing" suffix is in order here)

Line 110: and demonstrated (not "has"; please maintain simple past tense)

Line 205: not determined (correct, please!)

Line 212: Molina's ("s" missing)

Line 280: Diketopiperazines (is misspelled in the text)

Table 2: you could add a fourth column titled "References" to improve visual appearance

Conclusions: right now, these are just a summary of the review paper right now; please mention also some current gaps in knowledge and how you envision they could be filled by future research efforts

Comments on the Quality of English Language

There are quite a lot of misspellings and a few tense inconsistencies. Please re-read carefully the entire manuscript before resubmitting.

Author Response

In first place, I sincerely appreciate the valuable feedback provided to improve the quality of the paper. The reviewers’ insights are greatly valued, and I am committed to incorporating them effectively into the manuscript. In this sense, all the spelling mistakes and tense inconsistencies have been diligently corrected. As for the remaining comments and suggestions, I will address each point individually in separate responses.

- “This review summarizes the structural diversity, synthetic (or semi-synthetic) strategies for obtaining such chemical structures, and some of their biomedical applications. I believe it is a good, comprehensive overview of the state-of-the-art, but it could be better structured. Right now, it consists of three very disproportionate chapters. A reasonably-sized introduction, a short conclusion and a very large chapter 2, which is the bulk of the review. I would have preferred to see a more balanced distribution of material across different chapters, as announced in the Abstract: first an overview of structural classes, then synthetic methodologies, and finally therapeutic applications. This way the review will be easily perused by both organic chemists interested likely more in the former and biologists/biochemists/pharmacologists interested rather more in the latter aspects”.

Response: We value your input and have carefully considered your suggestion. In this sense, we shuffle at first about several approaches for the structure of the review, and finally decided to disclose jointly the chemical structure and the therapeutic application. The reason behind this idea was evidencing more clearly the structure/activity relationship. On the other hand, we have placed the synthetic strategies separately and at the end of each family to draw the attention of organic chemists in a more specific way.

 -“As a comprehensive review, I was surprised to find that staurosporine, a promising anticancer compound from this class, was not mentioned. See, for example:

 https://www.frontiersin.org/journals/microbiology/articles/10.3389/fmicb.2022.957473/full, or other similar literature sources”.Also, tryptophol, a sleep-inducing compound isolated from marine sponge Ircinia spiculosa could have been mentioned as well”.

Response: We completely agree with this comment and deeply apologise for missing out this important information. Tryptophol and Staurosporine have now been included in the manuscript within their corresponding section (2.1.1, p. 4 and 2.3.4, p. 45, respectively), and the rest of the structures have been renumbered accordingly.

-“Also, although I am not sure they have been discovered in marine species to date, I wonder if any secodine alkaloids could be included also as part of the structural diversity outlined in the review. Consider adding these as well if any recent literature on marine sources exists”.

Response: In addressing your inquiry, secodine alkaloids are indeed documented in the literature as Monoterpene Indole Alkaloids (MIAs) primarily sourced from Apocynaceae Genera. However, we have not come across any references indicating the presence of secodine alkaloids sourced from marine organisms. Thus, while we acknowledge their existence within terrestrial plants, their inclusion in our review, given the lack of evidence of marine sources, may not be warranted.

- Other minor issues that have been addressed:

-Title has been changed: Current Status of Indole-derived Marine Natural Products: Synthetic Approaches and Therapeutic Applications”

-Line 5: “London” has been corrected

-Line 4: “comma needed before "Daniel" has been added: “Sergio Fernández 1, Virginia Arnaiz 2, Daniel Rufo 2 and Yolanda Arroyo 3,*”

-Line 7: “Science” has been corrected

-Line 26: “were shown” has been corrected

-Line 42: The sentence has been changed to “with indole derivatives being one the most promising”

-Line 66: indole, spelling issue has been corrected

-Line 94: consist ("-ing" suffix has been removed)

-Line 110: and demonstrated (past tense has been corrected)

-Line 205: “not determined” has been corrected)

-Line 212: Molina's ("s" has been added)

-Line 280: “Diketopiperazine” has been corrected

-Table 2: fourth column titled "References" has been added

-“Conclusions: right now, these are just a summary of the review paper right now; please mention also some current gaps in knowledge and how you envision they could be filled by future research efforts”

Response: Thank you for your feedback. We have revised the Conclusions section to incorporate not only a summary of the review paper but also to highlight current gaps in knowledge. Additionally, we discuss potential avenues for future research efforts aimed at addressing these gaps. We believe this enhancement provides a more comprehensive and forward-looking perspective to the conclusions of the paper.